# Glycan-based shaping of the microbiota during primate evolution

**Sumnima Singh[‡], Patricia Bastos-Amador[†§], Jessica Ann Thompson[†], Mauro Truglio, Bahtiyar Yilmaz[#], Silvia Cardoso, Daniel Sobral[¶], Miguel P Soares***

Instituto Gulbenkian de Ciência, Oeiras, Portugal

**Abstract** Genes encoding glycosyltransferases can be under relatively high selection pressure, likely due to the involvement of the glycans synthesized in host-microbe interactions. Here, we used mice as an experimental model system to investigate whether loss of $\alpha-1,3$-galactosyltransferase gene (*GGTA1*) function and Gal$\alpha$1-3Gal$\beta$1-4GlcNAc$\beta$1-R ($\alpha$Gal) glycan expression affects host-microbiota interactions, as might have occurred during primate evolution. We found that *Ggta1* deletion shaped the composition of the gut microbiota. This occurred via an immunoglobulin (Ig)-dependent mechanism, associated with targeting of $\alpha$Gal-expressing bacteria by IgA. Systemic infection with an Ig-shaped microbiota inoculum elicited a less severe form of sepsis compared to infection with non-Ig-shaped microbiota. This suggests that in the absence of host $\alpha$Gal, antibodies can shape the microbiota towards lower pathogenicity. Given the fitness cost imposed by bacterial sepsis, we infer that the observed reduction in microbiota pathogenicity upon *Ggta1* deletion in mice may have contributed to increase the frequency of *GGTA1* loss-of-function mutations in ancestral primates that gave rise to humans.

**\*For correspondence:**
mpsoares@igc.gulbenkian.pt

[†]These authors contributed equally to this work

**Present address:** [‡]Center for Systems Biology, Massachusetts General Hospital and Harvard Medical School, Boston, United States; [§]Champalimaud Centre for the Unknown, Lisbon, Portugal; [#]Department for Biomedical Research, Inselspital, University of Bern, Bern, Switzerland; [¶]Universidade Nova de Lisboa, Caparica, Portugal

**Competing interests:** The authors declare that no competing interests exist.

## Introduction

As originally proposed by J.B.S. Haldane, infectious diseases are a major driving force of natural selection (*Haldane, 1949*), occasionally precipitating 'catastrophic-selection' events: the replacement of an entire, susceptible, parental population by mutant offspring that are resistant to a given infectious disease (*Lewis, 1962*). Such an event is proposed to have occurred during primate evolution between 20–30 million-years-ago, possibly due to the selective pressure exerted by an airborne, enveloped virus carrying Gal$\alpha$1-3Gal$\beta$1-4GlcNAc ($\alpha$Gal)-like glycans (*Galili, 2016*; *Galili, 2019*). If proven correct, this would contribute to the evolutionary pressure that led to the selection and fixation of *GGTA1* loss-of-function mutations in ancestral primates (*Galili et al., 1988*).

Loss of the $\alpha$1,3-galactosyltransferase enzyme, encoded by *GGTA1*, eliminated the expression of protein-bound $\alpha$Gal, allowing for immune targeting of this non-self glycan (*Galili et al., 1987*). This increased resistance to infection by $\alpha$Gal-expressing pathogens (*Repik et al., 1994*; *Takeuchi et al., 1996*), including parasites of the *Plasmodium* spp. (*Soares and Yilmaz, 2016*; *Yilmaz et al., 2014*), which exerted a major impact on human evolution (*Allison, 1954*).

We recently uncovered a possible fitness advantage associated with loss of *GGTA1* function, which acts independently of $\alpha$Gal-specific immunity (*Singh et al., 2021*). Namely, loss of $\alpha$Gal from immunoglobulin (Ig)G-associated glycan structures increased IgG effector function and resistance to bacterial sepsis in mice (*Singh et al., 2021*).

Sepsis is a life-threatening organ dysfunction caused by a deregulated response to infection (*Singer et al., 2016*) that accounts for 20% of global human mortality (*Rudd et al., 2020*). The pathogenesis of sepsis is modulated by stable host symbiotic associations with microbial communities composed of bacteria, fungi, and viruses, known as the microbiota (*Rudd et al., 2020*; *Vincent et al., 2009*). While host-microbiota interactions provide a broad range of fitness advantages to the host (*Lane-Petter, 1962*; *Vonaesch et al., 2018*), these also carry fitness costs, for

example, when bacterial pathobionts (*Chow et al., 2011*) translocate across host epithelial barriers to elicit the development of sepsis (*Rudd et al., 2020*; *Vincent et al., 2009*). On the basis of this evolutionary trade-off (*Stearns and Medzhitov, 2015*), it has been argued that the immune system may have emerged, in part, to mitigate the pathogenic effects of the microbiota (*Hooper et al., 2012*; *McFall-Ngai, 2007*). Central to this host defence strategy is the transepithelial secretion of copious amounts of IgA natural antibodies (NAb), which target immunogenic bacteria in the microbiota (*Macpherson et al., 2000*).

IgA recognize a broad but defined subset of immunogenic bacteria in the gut microbiota (*Bunker et al., 2017*; *Bunker et al., 2015*; *Macpherson et al., 2000*), exerting negative or positive selection pressure on these bacteria, shaping the microbiota composition, ecology, and potentially its pathogenicity (*Kubinak and Round, 2016*). Negative selection can occur, for example, when IgA limits bacterial growth (*Moor et al., 2017*), while positive selection can occur, for example, when IgA promotes bacterial interactions with the host, favoring bacterial retention, fitness, and colonization (*Donaldson et al., 2018*; *McLoughlin et al., 2016*). Moreover, IgA can interfere with cognate interactions between bacteria and tissue resident immune cells at epithelial barriers, regulating microbiota-specific immune responses, including the production of circulating IgM and IgG NAb (*Kamada et al., 2015*; *Zeng et al., 2016*).

Here, we provide experimental evidence in mice to suggest that the fixation of *GGTA1* loss-of-function mutations during primate evolution exerted a major impact on the composition of their gut microbiota. In support of this notion, mice in which *Ggta1* is disrupted (*Ggta1$^{-/-}$*), mimicking human *GGTA1* loss-of-function mutations, modulated their gut microbiota composition. This occurs predominantly via an Ig-dependent mechanism, associated with an enhancement of the production of IgA, targeting αGal-expressing bacteria in the gut microbiota. The pathogenicity of the Ig-shaped microbiota is reduced, failing to elicit lethal forms of sepsis upon systemic infection. We propose that *GGTA1* loss-of-function mutations conferred a selective benefit during primate evolution, in part, by shaping commensal bacteria in the microbiota to mitigate the pathogenesis of sepsis.

## Results

### *Ggta1* deletion shapes the microbiota composition

We have previously established that *Ggta1$^{-/-}$* mice harbor a distinct microbiota composition to that of wild type (*Ggta1$^{+/+}$*) mice (*Singh et al., 2021*). This is illustrated by the relative abundance of specific bacterial taxa, such as an increase in Proteobacteria, Tenericutes, and Verrucomicrobia as well as a reduction in Bacteroidetes and Deferribacteres phyla in *Ggta1$^{-/-}$* mice compared to *Ggta1$^{+/+}$* mice (*Figure 1A*, *Figure 1—figure supplements 1* and *2*; *Singh et al., 2021*). The relative increase of Proteobacteria, a phylum containing several strains associated with pathogenic behavior, in the gut microbiota of *Ggta1$^{-/-}$* mice was not, however, associated with the development of histological lesions in the gastrointestinal tract (*Figure 1—figure supplement 3A*). Absence of intestinal inflammation was further assessed by quantification of fecal lipocalin-2 (Lcn-2) (*Figure 1—figure supplement 3B*; *Chassaing et al., 2012*). There were also no histopathological lesions in the liver, lungs, kidney, and spleen (*Figure 1—figure supplement 3C*), suggesting that *Ggta1$^{-/-}$* mice maintain symbiotic interactions with these pathobionts, without compromising organismal homeostasis.

To establish whether the differences in the bacterial species present in the gut microbiota of *Ggta1$^{-/-}$ vs. Ggta1$^{+/+}$* mice are propelled by host genetics, we used an experimental system whereby the microbiota is vertically transmitted over several generations (*Ubeda et al., 2012*) from *Ggta1$^{+/+}$* mice to *Ggta1$^{-/-}$* and *Ggta1$^{+/+}$* offspring (*Figure 1B*). This approach enables effects exerted by the host genotype on microbiota composition to predominate over those exerted by environmental factors (*Gálvez et al., 2017*; *Vonaesch et al., 2018*), diet (*Sonnenburg et al., 2016*), cohousing or familial transmission (*Ubeda et al., 2012*), albeit not accounting for putative cage effects or genetic drift (*Spor et al., 2011*).

The microbiota composition of F$_1$ *Ggta1$^{+/-}$* well as F$_2$ *Ggta1$^{+/+}$* and *Ggta1$^{-/-}$* mice diverged from that of the original F$_0$ *Ggta1$^{+/+}$* mice (*Figure 1C*). While indistinguishable in F$_2$ *Ggta1$^{+/+}$* and *Ggta1$^{-/-}$* littermates (*Figure 1C*; *Singh et al., 2021*), the microbiota composition diverged between *Ggta1$^{+/+}$* and *Ggta1$^{-/-}$* mice in F$_3$ (*Figure 1D*), F$_4$ (*Figure 1E*) and F$_5$ (*Figure 1F*) generations, suggesting that the *Ggta1* genotype per se alters microbiota composition. There was an enrichment of

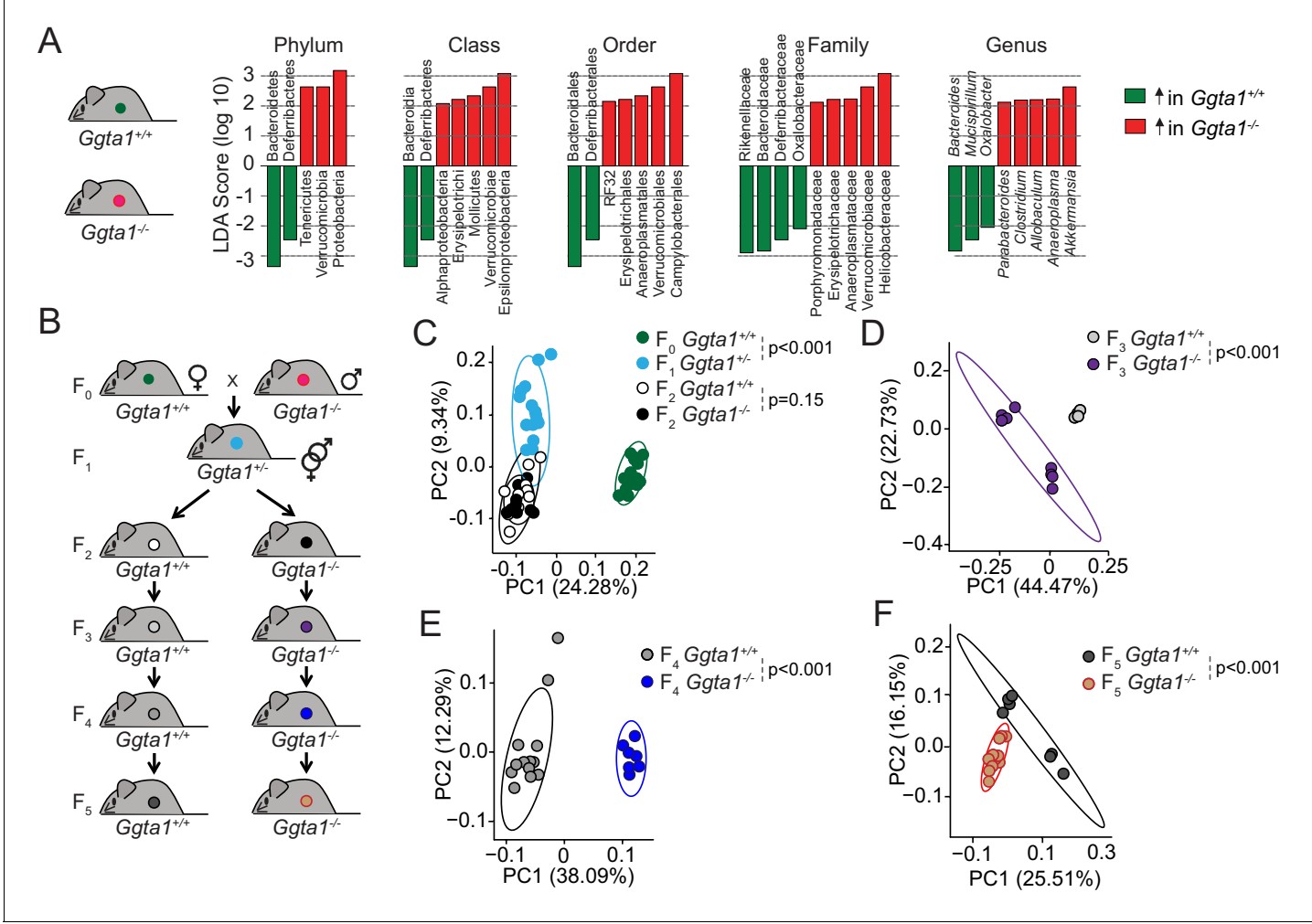

**Figure 1.** *Ggta1* deletion alters the gut microbiota. (A) Linear discriminant analysis (LDA) scores generated from LEfSe analysis, representing taxa enriched in the fecal microbiota of *Ggta1^{+/+}^* (green) (n = 15) and *Ggta1^{-/-}^* (red) (n = 14) mice. (B) Breeding strategy where F$_0$ *Ggta1^{-/-}^* males were crossed with *Ggta1^{+/+}^* females to generate F$_1$ *Ggta1^{+/-}^* mice, which were bred to generate F$_2$ *Ggta1^{+/+}^* vs. *Ggta1^{-/-}^* littermates. These were subsequently bred to generate F$_3$ to F$_5$ *Ggta1^{+/+}^* vs. *Ggta1^{-/-}^* mice. Microbiota Principal Coordinate Analysis of Unweighted Unifrac distance in fecal samples from (C) F$_0$ *Ggta1^{+/+}^* (n = 15), F$_1$ *Ggta1^{+/-}^* (n = 15), F$_2$ *Ggta1^{+/+}^* (n = 11) and F$_2$ *Ggta1^{-/-}^* (n = 10) mice, (D) F$_3$ *Ggta1^{+/+}^* (n = 9) and *Ggta1^{-/-}^* (n = 8) mice, (E) F$_4$ *Ggta1^{+/+}^* (n = 13) and *Ggta1^{-/-}^* (n = 7) mice and (F) F$_5$ *Ggta1^{+/+}^* (n = 7) and *Ggta1^{-/-}^* (n = 12) mice generated as in (B). Data from one experiment with two to three independent breedings/cages per genotype. Symbols (C–F) are individual mice. p Values (C–F) calculated using PERMANOVA test.

The online version of this article includes the following figure supplement(s) for figure 1:

**Figure supplement 1.** Analyses of gut microbiota composition of *Ggta1^{+/+}^* and *Ggta1^{-/-}^* mice.

**Figure supplement 2.** Analyses of gut microbiota composition of *Ggta1^{+/+}^* vs. *Ggta1^{-/-}^* mice.

**Figure supplement 3.** *Ggta1* deletion does not cause inflammation at steady state.

**Figure supplement 4.** *Ggta1* deletion shapes the gut microbiota composition.

some bacterial taxa, such as Helicobactereaceae, in the microbiota of F$_2$ to F$_5$ *Ggta1^{-/-}^* and *Ggta1^{+/+}^* mice (*Figure 1—figure supplement 4*), despite the absence of these bacteria in the original F$_0$ *Ggta1^{+/+}^* microbiota (*Figure 1—figure supplement 4*). This suggests that the *Ggta1* genotype shapes the gut microbiota composition via a process that is probably influenced by colonization by environmental pathobionts (*Gálvez et al., 2017*).

### *Ggta1* deletion enhances IgA responses to the gut microbiota

Considering that IgA shapes the bacterial species present in the microbiota (*Bunker et al., 2015*; *Macpherson et al., 2018*; *Peterson et al., 2007*), we asked whether differences in IgA responses

could contribute to shape the microbiota of $Ggta1^{-/-}$ vs. $Ggta1^{+/+}$ mice. In keeping with this notion, the relative levels of microbiota-reactive circulating IgA were higher in $Ggta1^{-/-}$ vs. $Ggta1^{+/+}$ mice, maintained under specific pathogen-free (SPF) but not germ-free (GF) conditions (*Figure 2A*). Under SPF conditions, $Ggta1^{-/-}$ mice had similar levels of secreted (*Figure 2—figure supplement 1A*) and circulating (*Figure 2—figure supplement 1B*) total IgA, compared to $Ggta1^{+/+}$ mice. These were reduced to a similar extent under GF conditions (*Figure 2—figure supplement 1A and B*), suggesting that *Ggta1* deletion enhances microbiota-specific IgA responses without interfering with total IgA.

In further support of the notion that *Ggta1* deletion modulates the production of microbiota-specific IgA, colonization of GF $Ggta1^{-/-}$ mice with the microbiota from $Ggta1^{-/-}$ mice elicited the production of higher levels of microbiota-reactive circulating IgA, compared to GF $Ggta1^{+/+}$ mice colonized by the same microbiota (*Figure 2B*). This difference was not observed upon colonization of GF $Ggta1^{-/-}$ vs. $Ggta1^{+/+}$ mice by a microbiota isolated from $Ggta1^{+/+}$ mice (*Figure 2—figure supplement 1C*). This suggests that the enhanced microbiota-reactive IgA response of $Ggta1^{-/-}$ vs. $Ggta1^{+/+}$ mice is induced by immunogenic bacteria present specifically in the microbiota of $Ggta1^{-/-}$ mice.

Of note, the levels of circulating IgA were reduced in $Tcrb^{-/-}Ggta1^{-/-}$ mice lacking α/β T cells, when compared to $Ggta1^{-/-}$ mice (*Figure 2—figure supplement 1D*), suggesting that the production of circulating IgA NAb in $Ggta1^{-/-}$ mice occurs, in part, via a T-cell-dependent mechanism, which is consistent with previous reports in $Ggta1^{+/+}$ mice (*Bunker et al., 2017*; *Fagarasan et al., 2010*; *Macpherson et al., 2000*).

We then asked whether $Ggta1^{-/-}$ mice shape their microbiota via a mechanism associated with immune targeting of αGal-expressing bacteria. Consistent with a number of bacteria in the human gut microbiota carrying genes orthologous to the mammalian α1,3-galactosyltransferase (*Montassier et al., 2019*), several human probiotic bacteria expressed αGal-like glycans at the cell surface when cultured in vitro (*Figure 2C*, *Figure 2—figure supplement 2*). Subsequent in vivo analyses demonstrated that approximately 20% of the bacteria in the small intestine of $Ggta1^{+/+}$ mice expressed αGal-like glycans at the cell surface (*Figure 2D,E*, *Figure 2—figure supplement 2*). Nearly 30% of these αGal+ bacteria were immunogenic (*Figure 2D,E*), as defined by the detection of surface-bound IgA (*Palm et al., 2014*), which predominantly targets bacteria in the small intestine (*Bunker et al., 2017*; *Bunker et al., 2015*). These IgA+αGal+ bacteria accounted for roughly 50% of all the immunogenic (IgA+) bacteria in the small intestine (*Figure 2D,E*). These were also present, although at a lower extent, in the cecum, colon, and feces (*Figure 2—figure supplement 1E and F*).

$Ggta1^{-/-}$ mice harbored a relatively lower percentage of immunogenic IgA+αGal+ bacteria in the small intestine, when compared to $Ggta1^{+/+}$ mice (*Figure 2D,E*), while the percentage of immunogenic IgA+ bacteria was similar in $Ggta1^{-/-}$ vs. $Ggta1^{+/+}$ mice (*Figure 2D,E*). This is consistent with the idea of a specific mechanism altering the microbiota of $Ggta1^{-/-}$ mice that presumably, at least in part, involves targeting immunogenic αGal+ bacteria by IgA. Whether this mechanism involves the recognition of bacterial αGal-like glycans by IgA is not clear.

We then compared the effect of IgA on the levels of systemic IgM and/or IgG NAb directed against antigens expressed by bacteria present in the microbiota (*Kamada et al., 2015*; *Zeng et al., 2016*). Induction of dysbiosis, by streptomycin, increased the levels of circulating αGal-specific IgM and IgG in $Igha^{-/-}Ggta1^{-/-}$ vs. $Igha^{+/+}Ggta1^{-/-}$ mice (*Figure 2F,G*). This illustrates again that the IgA response of $Ggta1^{-/-}$ mice is distinct from that of $Ggta1^{+/+}$ mice, reducing systemic IgM and IgG responses against antigens expressed by bacteria in the microbiota, as illustrated for αGal-like glycans. Presumably this occurs via a mechanism whereby IgA prevents αGal+ bacteria or bacterial products associated to αGal from translocating across epithelial barriers and inducing systemic immune responses against this glycan.

## *Ggta1* deletion shapes the gut microbiota via an antibody (Ig)-dependent mechanism

Maintenance of microbiota composition across generations is sustained via maternal IgG transfer to the offspring through the placenta during the fetal period, and maternal IgM, IgG and IgA transfer via lactation during the neonatal period (*Gensollen et al., 2016*; *Koch et al., 2016*). Over time, newborns generate IgA that target immunogenic bacteria in the microbiota (*McLoughlin et al., 2016*; *Moor et al., 2017*), shaping its composition throughout adult life (*Kawamoto et al., 2014*). We

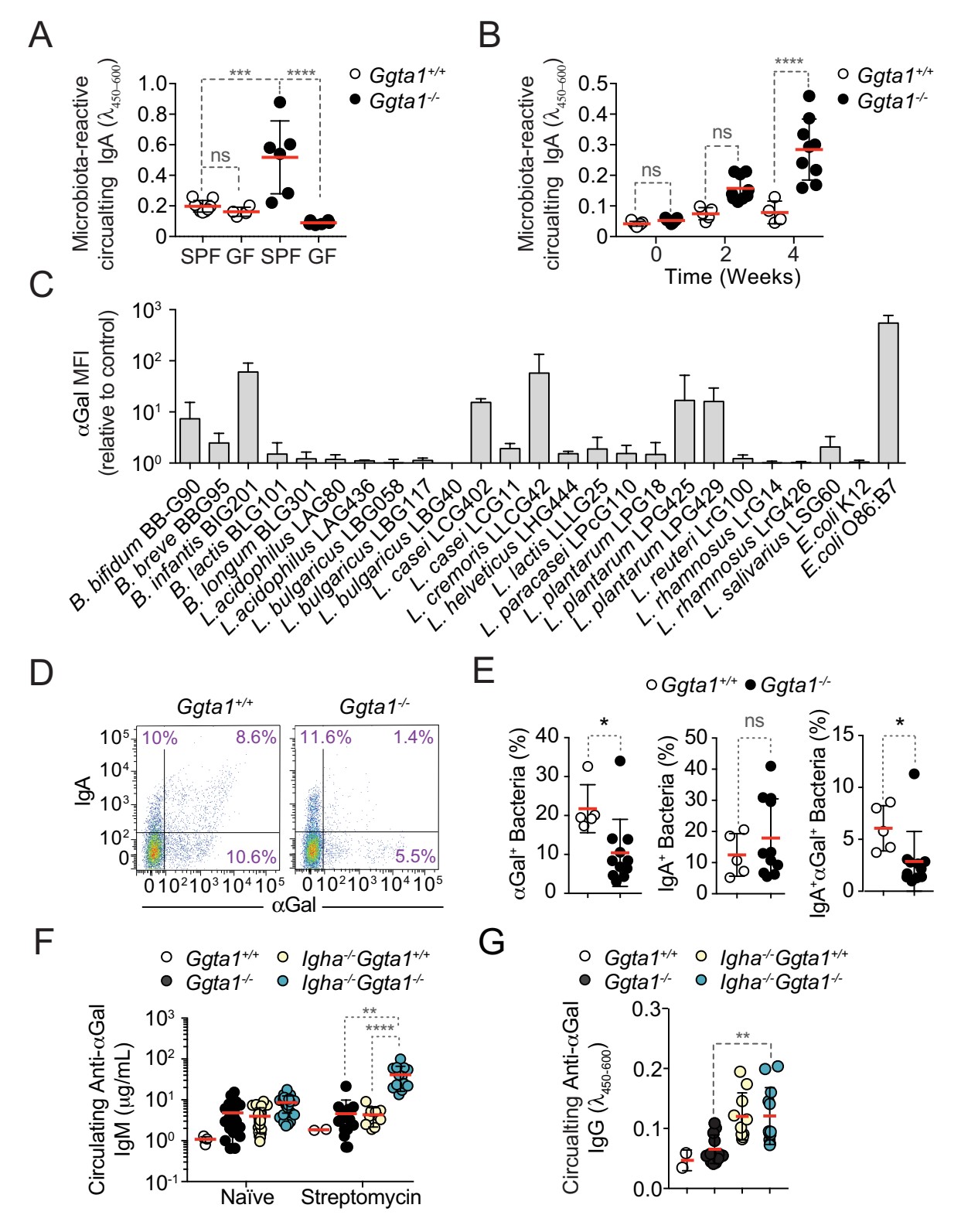

**Figure 2.** *Ggta1* deletion enhances IgA responses to the gut microbiota. (**A**) Relative binding of IgA in the serum of *Ggta1*<sup>+/+</sup> (n = 10) and GF *Ggta1*<sup>+/+</sup> (n = 5) mice to fecal extract from *Ggta1*<sup>+/+</sup> mice, and *Ggta1*<sup>-/-</sup> (n = 6) and GF *Ggta1*<sup>-/-</sup> (n = 6) mice to fecal extract from *Ggta1*<sup>-/-</sup> mice; one experiment. (**B**) Relative binding of IgA in the serum of GF *Ggta1*<sup>+/+</sup> (n = 7) and GF *Ggta1*<sup>-/-</sup> (n = 10) mice to cecal extract from *Ggta1*<sup>-/-</sup> mice at indicated time-points after colonization with cecal extract from *Ggta1*<sup>-/-</sup> mice; two experiments. (**C**) Median Fluorescence Intensity (MFI) of αGal<sup>+</sup> bacterial strains

*Figure 2 continued on next page*

*Figure 2 continued*

stained with BSI-B4 lectin relative to unstained control; seven experiments. (D) Representative flow cytometry plots showing bacteria stained for IgA and αGal in the small intestinal content of $Ggta1^{+/+}$ (n = 5) and $Ggta1^{-/-}$ (n = 11) mice; four independent experiments. (E) Quantification of αGal$^+$, IgA$^+$, and IgA$^+$αGal$^+$ bacteria in the same samples as in (D). (F) Concentration of anti-αGal IgM in serum of $Ggta1^{+/+}$ (n = 2), $Ggta1^{-/-}$ (n = 12), $Igha^{-/-}Ggta1^{+/+}$ (n = 10), and $Igha^{-/-}Ggta1^{-/-}$ (n = 12) mice before and after streptomycin treatment, two experiments. (G) Concentration of anti-αGal IgG, in the same mice as (F). Symbols (A, B, E, F, G) are individual mice. Red bars (A, B, E, F, G) correspond to mean values. Error bars (A, B, C, E, F, G) correspond to SD. p Values in (A, B, F, G) calculated using Kruskal-Wallis test using Dunn's multiple comparisons test and in (E) using Mann-Whitney test. *p<0.05, **p<0.01, ***p<0.001, ****p<0.0001, ns: not significant.

The online version of this article includes the following figure supplement(s) for figure 2:

**Figure supplement 1.** *Ggta1* deletion enhances IgA responses to the gut microbiota.
**Figure supplement 2.** αGal expression by probiotic bacteria.

hypothesized that the mechanism via which *Ggta1* deletion shapes the gut microbiota involves targeting of immunogenic bacteria by both maternally- and offspring-derived immunoglobulins (Ig). To test this hypothesis, we used a similar experimental approach to that described above (*Figure 1B*; *Ubeda et al., 2012*), whereby the microbiota from $Ggta1^{+/+}$ mice was vertically transmitted to $Ggta1^{+/+}$ or $Ggta1^{-/-}$ offspring that express Ig ($Igh\text{-}J^{+/+}$) or not ($Igh\text{-}J^{-/-}$) (*Figure 3A*). Crossing of $F_0$ $Igh\text{-}J^{+/+}Ggta1^{+/+}$ (female) with $Igh\text{-}J^{-/-}Ggta1^{-/-}$ (male) mice produced $F_1$ $Igh\text{-}J^{+/-}Ggta1^{+/-}$ mice, which were interbred to generate $F_2$ $Igh\text{-}J^{+/+}Ggta1^{+/+}$, $Igh\text{-}J^{+/+}Ggta1^{-/-}$, $Igh\text{-}J^{-/-}Ggta1^{+/+}$ and $Igh\text{-}J^{-/-}Ggta1^{-/-}$ mice, harboring a microbiota composition indistinguishable among the genotypes (*Figure 3—figure supplement 1A and B*). Consistent with our previous observations (*Figure 1C*; *Singh et al., 2021*), this suggests that maternal Ig transfer predominates over offspring Ig production in shaping the offspring microbiota, regardless of the genotype (*Ubeda et al., 2012*).

To dissect the relative contribution of the *Ggta1* from the *Ig* genotype in shaping the microbiota, $F_2$ littermates were interbred to generate $F_3$ offspring carrying a microbiota targeted by antibodies ($Igh\text{-}J^{+/+}Ggta1^{+/+}$ and $Igh\text{-}J^{+/+}Ggta1^{-/-}$) or not ($Igh\text{-}J^{-/-}Ggta1^{+/+}$ and $Igh\text{-}J^{-/-}Ggta1^{-/-}$) (*Figure 3A*). Consistent with our previous observations (*Figure 1*), there was a marked separation of the microbiota community structure between $F_3$ $Igh\text{-}J^{+/+}Ggta1^{-/-}$ *vs.* $Igh\text{-}J^{+/+}Ggta1^{+/+}$ mice, as assessed by Principal Coordinate Analyses (PCA) for Weighted and Unweighted Unifrac (*Figure 3B,C*). Considering that Weighted Unifrac accounts for the relative abundance of the bacterial taxa while Unweighted Unifrac accounts for only the presence or absence of the taxa in the microbial community (*Lozupone and Knight, 2005*), they represent differences exerted by high and low abundant bacterial taxa respectively. This suggests that *Ggta1* deletion shapes both high and low abundant taxa present in the microbiota when maternal and/or offspring-derived antibodies (i.e. Ig) are present.

LefSe analysis showed that the microbiota from $Igh\text{-}J^{+/+}Ggta1^{-/-}$ mice was enriched in specific bacterial taxa, including Proteobacteria, as compared to the microbiota from $Igh\text{-}J^{+/+}Ggta1^{+/+}$ mice (*Figure 3—figure supplement 1C*). This suggests that *Ggta1* deletion favors gut colonization by pathobionts in an Ig-dependent manner.

We then asked whether Ig are functionally involved in shaping the microbiota composition of $Ggta1^{-/-}$ *vs.* $Ggta1^{+/+}$ mice. In the absence of Ig ($Igh\text{-}J^{-/-}$), there were no differences in the microbiota composition of $F_3$ $Igh\text{-}J^{-/-}Ggta1^{-/-}$ *vs.* $Igh\text{-}J^{-/-}Ggta1^{+/+}$ mice, as assessed by PCA for Weighted Unifrac (*Figure 3B*). This suggests that shaping of highly abundant taxa in the microbiota of $Ggta1^{-/-}$ *vs.* $Ggta1^{+/+}$ mice occurs via an Ig-dependent mechanism. In contrast, the microbiota composition of $F_3$ $Igh\text{-}J^{-/-}Ggta1^{-/-}$ mice remained distinct from that of $Igh\text{-}J^{-/-}Ggta1^{+/+}$ mice, as assessed by PCA for Unweighted Unifrac (*Figure 3C*). This suggests that shaping of low abundance bacterial taxa in the microbiota of $Ggta1^{-/-}$ *vs.* $Ggta1^{+/+}$ mice occurs in an Ig-independent manner.

To address the extent to which *Ggta1* deletion promotes shaping of the microbiota via an Ig-dependent mechanism, we compared the Unifrac distances between $Igh\text{-}J^{+/+}Ggta1^{+/+}$ and $Igh\text{-}J^{-/-}Ggta1^{+/+}$ mice *vs.* $Igh\text{-}J^{+/+}Ggta1^{-/-}$ and $Igh\text{-}J^{-/-}Ggta1^{-/-}$ mice, similar to what is previously described (*Lozupone and Knight, 2005*). The Weighted Unifrac distance of microbiota from $Igh\text{-}J^{+/+}Ggta1^{-/-}$ *vs.* $Igh\text{-}J^{-/-}Ggta1^{-/-}$ mice was higher than that from $Igh\text{-}J^{+/+}Ggta1^{+/+}$ *vs.* $Igh\text{-}J^{-/-}Ggta1^{+/+}$ mice (*Figure 3D*). This suggests that the relative impact of Ig on the gut microbiota community structure exerted by highly abundant bacterial taxa is enhanced in $Ggta1^{-/-}$ *vs.* $Ggta1^{+/+}$ mice. This was confirmed by LefSe analysis showing an enhanced Ig-dependent enrichment of several bacterial

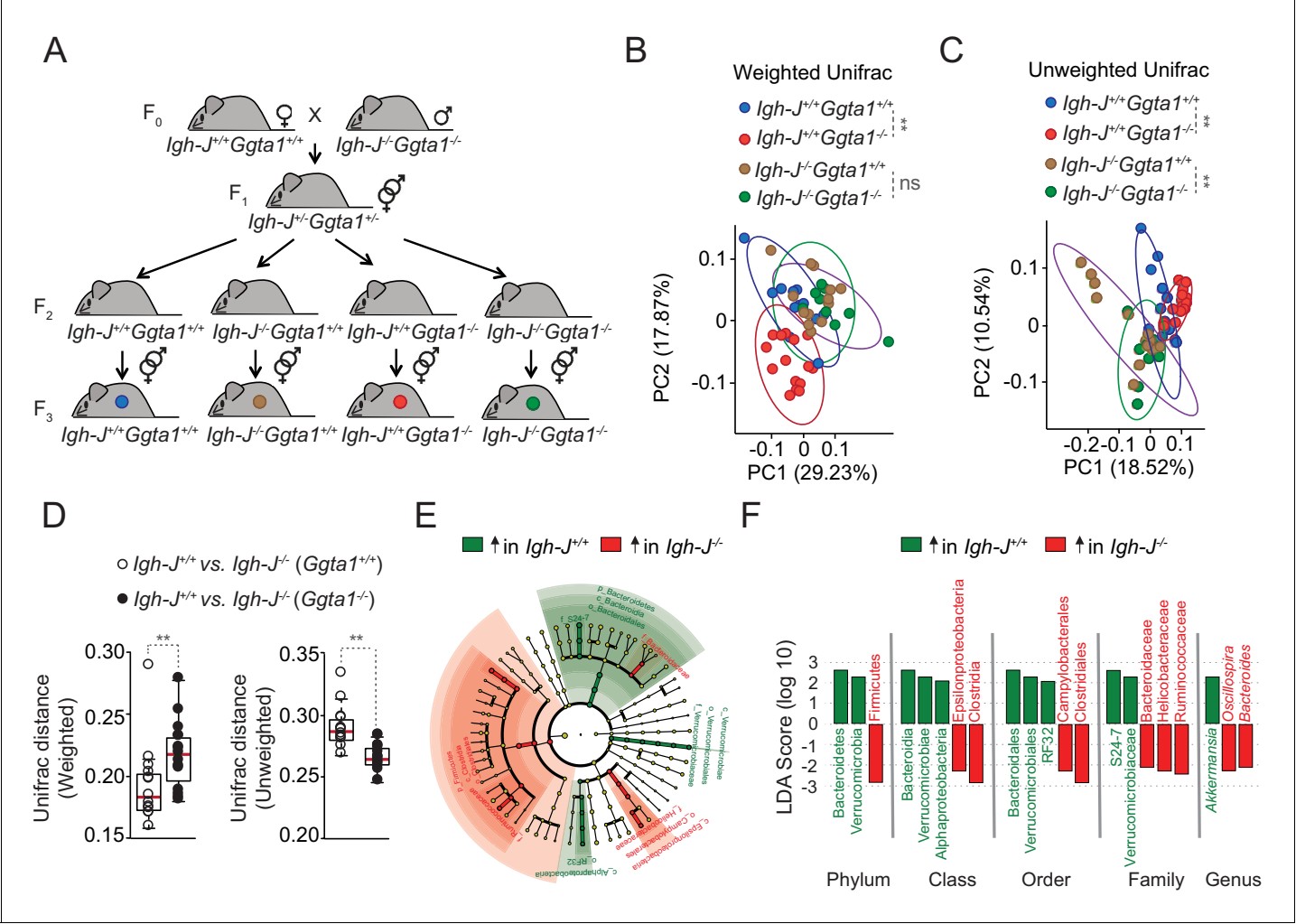

**Figure 3.** *Ggta1* deletion shapes the gut microbiota via an Ig-dependent mechanism. (**A**) Breeding strategy where F$_0$ *Igh-J$^{-/-}$Ggta1$^{-/-}$* males were crossed with *Igh-J$^{+/+}$Ggta1$^{+/+}$* females to generate F$_1$ *Igh-J$^{+/-}$Ggta1$^{+/-}$*mice, which were bred to generate F$_2$ and F$_3$ *Igh-J$^{+/+}$Ggta1$^{+/+}$*, *Igh-J$^{-/-}$Ggta1$^{+/+}$*, *Igh-J$^{+/+}$Ggta1$^{-/-}$, and Igh-J$^{-/-}$Ggta1$^{-/-}$* mice. Microbiota Principal Coordinate Analysis of (**B**) Weighted and (**C**) Unweighted Unifrac and (**D**) Distance of Weighted and Unweighted Unifrac of 16S rRNA amplicons, in fecal samples from F$_3$ *Igh-J$^{+/+}$Ggta1$^{+/+}$* (n = 13) *vs.* *Igh-J$^{-/-}$Ggta1$^{+/+}$* (n = 16) mice and F$_3$ *Igh-J$^{+/+}$Ggta1$^{-/-}$* (n = 13) *vs.* *Igh-J$^{-/-}$Ggta1$^{-/-}$* (n = 11) mice generated as in (**A**). (**E**) Cladogram and (**F**) Linear discriminant analysis (LDA) scores generated from LEfSe analysis, representing taxa enriched in the fecal microbiota of the same mice as in (**B–D**). Data from one experiment with two to three independent breedings/cages per genotype. Symbols (**B, C, D**) are individual mice. Red bars (**D**) correspond to mean values. Error bars (**D**) correspond to SD. p Values in (**B, C**) calculated using PERMANOVA and in (**D**) using Mann-Whitney test. \*\*p<0.01, ns: not significant.

The online version of this article includes the following figure supplement(s) for figure 3:

**Figure supplement 1.** *Ggta1* deletion shapes the gut microbiota via an Ig-dependent mechanism.

taxa in the microbiota of *Ggta1$^{-/-}$* mice (***Figure 3E–F***), compared to that of *Ggta1$^{+/+}$* mice (***Figure 3—figure supplement 1D***). In contrast, the Unweighted Unifrac distance of *Igh-J$^{+/+}$Ggta1$^{+/+}$* vs. *Igh-J$^{-/-}$Ggta1$^{+/+}$* mice was higher compared to *Igh-J$^{+/+}$Ggta1$^{-/-}$* vs. *Igh-J$^{-/-}$Ggta1$^{-/-}$* mice (***Figure 3D***). This suggests that the relative impact of Ig on the microbiota community structure of low abundant bacterial taxa is less pronounced for *Ggta1$^{-/-}$* vs. *Ggta1$^{+/+}$* mice.

We then asked whether Ig exert a higher impact on the relative abundance of pathobionts in the gut microbiota of *Ggta1$^{-/-}$* vs. *Ggta1$^{+/+}$* mice. In strong support of this notion, the gut microbiota from *Igh-J$^{-/-}$Ggta1$^{-/-}$* mice, lacking Ig, was enriched with Helicobactereaceae family from Proteobacteria phylum as compared to *Igh-J$^{+/+}$Ggta1$^{-/-}$* mice expressing Ig (***Figure 3E,F***). This is consistent with our previous finding that the gut microbiota of *Rag2$^{-/-}$Ggta1$^{-/-}$* mice, lacking adaptive immunity,

is highly enriched in Proteobacteria, including *Helicobacter* (*Singh et al., 2021*). Of note, this was not observed in *Igh-J$^{-/-}$Ggta1$^{+/+}$ vs. Igh-J$^{+/+}$Ggta1$^{+/+}$* mice (*Figure 3—figure supplement 1D*). These data suggest that the absence of host αGal favors colonization of the gut microbiota by pathobionts, the expansion of which is restrained by Ig.

### *Ggta1* deletion reduces microbiota pathogenicity

We then asked whether the Ig-dependent shaping of the microbiota in *Ggta1$^{-/-}$* mice affects the pathogenesis of sepsis due to systemic infections emanating from gut microbes. *Ggta1$^{-/-}$* mice were protected against systemic infections (i.p.) by a cecal inoculum isolated from *Rag2$^{-/-}$Ggta1$^{-/-}$* mice, reflecting a microbiota not shaped by Ig (*Figure 4—figure supplement 1A and B*). This is consistent with our previous finding showing *Ggta1* deletion enhances protection against systemic bacterial infections (i.p.) via a mechanism involving IgG NAb (*Singh et al., 2021*). Moreover, *Ggta1$^{-/-}$* mice were also protected against a systemic infection (i.p.) by a cecal inoculum isolated from *Ggta1$^{-/-}$* mice, reflecting their own Ig-shaped microbiota (*Figure 4—figure supplement 1A and B*). This is in keeping with the previously shown enhanced protection of *Ggta1$^{-/-}$* mice against cecal ligation and puncture (*Singh et al., 2021*). Surprisingly *Igh-J$^{-/-}$Ggta1$^{-/-}$* mice lacking B cells (*Figure 4—figure supplement 1C*), *Tcrb$^{-/-}$Ggta1$^{-/-}$* mice lacking α/β T cells (*Figure 4—figure supplement 1D*) and *Rag2$^{-/-}$Ggta1$^{-/-}$* mice lacking B and T cells (*Figure 4—figure supplement 1E*) remained protected against systemic infections by the cecal inoculum isolated from *Ggta1$^{-/-}$* mice. This suggests that the previously described IgG-dependent mechanism that protects *Ggta1$^{-/-}$* mice from a systemic infection by a cecal inoculum isolated from *Rag2$^{-/-}$Ggta1$^{-/-}$* mice (*Singh et al., 2021*) is distinct from that protecting *Ggta1$^{-/-}$* mice from a systemic infection by their own cecal inoculum. We reasoned that this might be explained by a reduction of the overall pathogenicity of the Ig-shaped microbiota of *Ggta1$^{-/-}$* mice, compared to the non-Ig-shaped microbiota of *Rag2$^{-/-}$Ggta1$^{-/-}$* mice. To test this hypothesis, we compared the outcome of *Rag2$^{-/-}$Ggta1$^{-/-}$* mice upon a systemic infection by Ig-shaped *vs.* a non-Ig-shaped microbiota.

*Rag2$^{-/-}$Ggta1$^{-/-}$* mice remained protected against systemic infections by the cecal inoculum isolated from *Ggta1$^{-/-}$* mice, while succumbing to a systemic infection by a cecal inoculum isolated from *Rag2$^{-/-}$Ggta1$^{-/-}$* mice (*Figure 4A,B*). Lethality was associated with a $10^6$–$10^7$-fold increase in bacterial load (*Figure 4C*), suggesting that the Ig-shaped microbiota of *Ggta1$^{-/-}$* mice is less pathogenic, when compared to the non-Ig-shaped microbiota from *Rag2$^{-/-}$Ggta1$^{-/-}$* mice.

We then asked whether the reduction of microbiota pathogenicity imposed by the adaptive immune system of *Ggta1$^{-/-}$* mice is also operational in *Ggta1$^{+/+}$* mice. However, *Rag2$^{-/-}$Ggta1$^{+/+}$* mice succumbed to the same extent to systemic infection (i.p.) with a cecal inoculum isolated from either *Rag2$^{+/+}$Ggta1$^{+/+}$ vs. Rag2$^{-/-}$Ggta1$^{+/+}$* mice (*Figure 4D–E*), developing similar bacterial loads (*Figure 4F*). This suggests that the mechanism via which the adaptive immune system of *Ggta1$^{-/-}$* mice shapes and reduces the pathogenicity of its microbiota is not operational in *Ggta1$^{+/+}$* mice.

Having established that in the absence of adaptive immunity, *Ggta1$^{-/-}$* mice are resistant against systemic infection by a low pathogenic Ig-shaped microbiota (*Figure 4—figure supplement 1C and E*), we asked whether the mechanism of resistance relies on the innate immune system. Depletion of Ly6C$^+$/Ly6G$^+$ myeloid cells (i.e. polymorphonuclear cells and inflammatory monocytes) using an anti-GR1 monoclonal Ab (*Figure 4—figure supplement 1F*; *Daley et al., 2008*) increased the susceptibility of *Ggta1$^{-/-}$* mice to systemic infection by their cecal inoculum (*Figure 4G*). This was associated with a $10^2$–$10^5$-fold increase in bacterial load, compared to control *Ggta1$^{-/-}$* mice (*Figure 4H*). Of note, monocyte/macrophage depletion by Clodronate liposomes (*Figure 4—figure supplement 1G*; *Sunderkötter et al., 2004*) failed to compromise the survival of *Ggta1$^{-/-}$* mice upon infection by the same cecal inoculum (*Figure 4—figure supplement 1H*). This suggests that polymorphonuclear cells are essential for resistance against systemic infection emanating from the less pathogenic Ig-shaped microbiota of *Ggta1$^{-/-}$* mice.

## Discussion

While loss-of-function mutations in genes encoding glycosyltransferases can provide fitness advantages against infection, these can compromise the physiologic functions of the eliminated self-glycan, as illustrated by the occurrence of reproductive senescence upon *Ggta1* deletion in mice (*Singh et al., 2021*). In an evolutionary context, such a trade-off could explain why loss-of-function

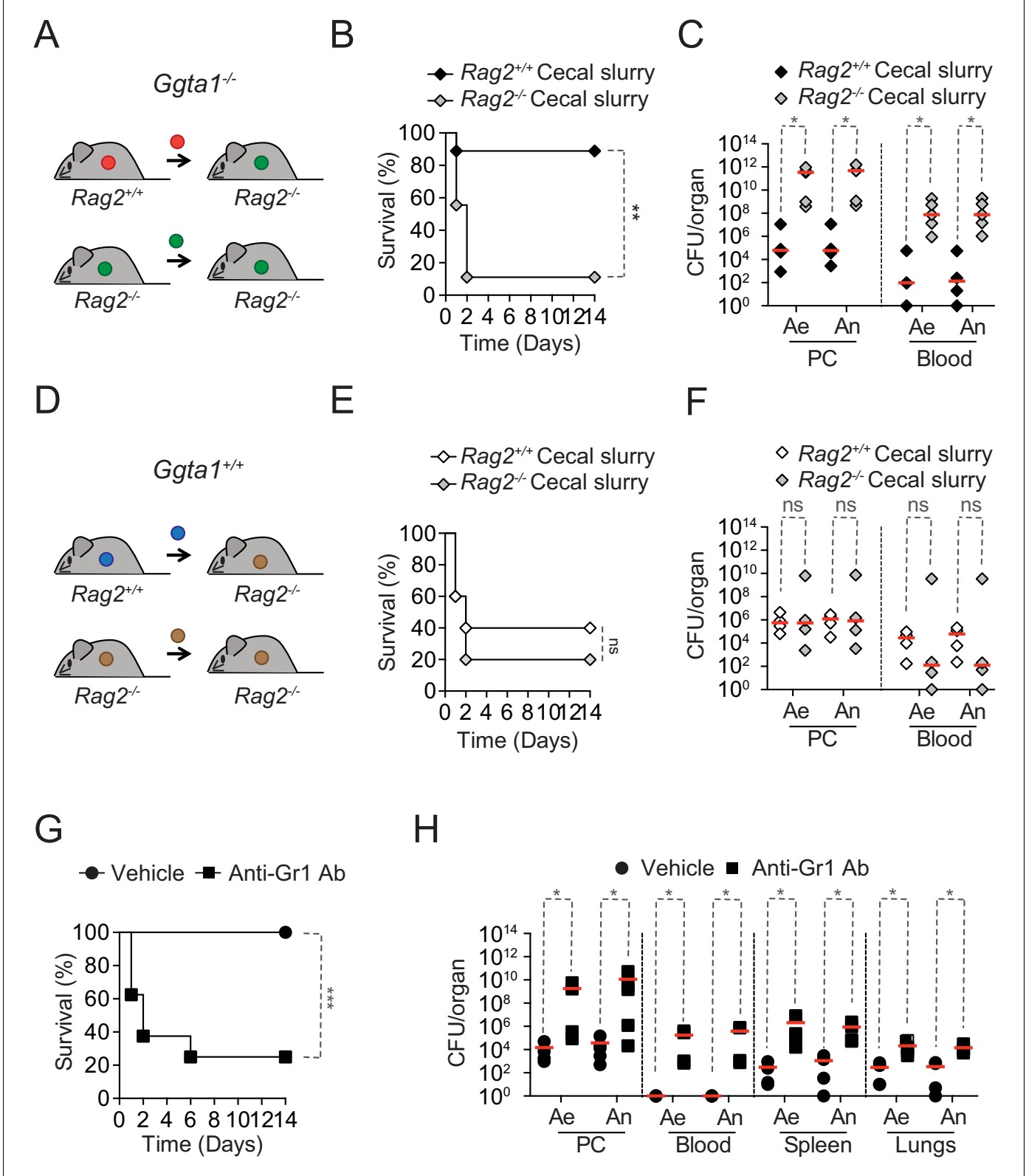

**Figure 4.** *Ggta1* deletion reduces microbiota pathogenicity. (**A**) Schematic showing infection of *Rag2⁻/⁻Ggta1⁻/⁻* mice with a cecal inoculum from either *Ggta1⁻/⁻* or *Rag2⁻/⁻Ggta1⁻/⁻* mice. (**B**) Survival of *Rag2⁻/⁻Ggta1⁻/⁻* (n = 9) mice after infection with a cecal inoculum from *Ggta1⁻/⁻* mice, and from *Rag2⁻/⁻Ggta1⁻/⁻* (n = 9) mice after infection with a cecal inoculum from *Rag2⁻/⁻Ggta1⁻/⁻* mice; two experiments. (**C**) Colony-forming units (CFU) of aerobic (Ae) and anaerobic (An) bacteria in *Rag2⁻/⁻Ggta1⁻/⁻* mice (n = 5 per group), 24 hr after infection as in (**B**); two experiments. (**D**) Schematic showing

*Figure 4 continued on next page*

Figure 4 continued

infection of $Rag2^{-/-}Ggta1^{+/+}$ mice with a cecal inoculum from either $Ggta1^{+/+}$ or $Rag2^{-/-}Ggta1^{+/+}$ mice. (E) Survival of $Rag2^{-/-}Ggta1^{+/+}$ (n = 5) mice after infection with a cecal inoculum from $Ggta1^{+/+}$ mice, and $Rag2^{-/-}Ggta1^{+/+}$ (n = 5) mice after infection with a cecal inoculum from $Rag2^{-/-}Ggta1^{+/+}$ mice; one experiment. (F) CFU of Ae and An bacteria in $Rag2^{-/-}Ggta1^{-/-}$ mice (n = 4 per group), 24 hr after infection as in (E); one experiment. (G) Survival of $Ggta1^{-/-}$ mice receiving vehicle (PBS) (n = 9) or Anti-Gr1 Ab (n = 8), 24 hr before infection with cecal inoculum from $Ggta1^{-/-}$ mice; two experiments. (H) CFU of Ae and An bacteria in $Ggta1^{-/-}$ mice receiving vehicle (PBS) (n = 4–5) or Anti-Gr1 Ab (n = 4–5), 24 hr after infection as in (G); five experiments. Symbols (C, F, H) represent individual mice. Red lines (C, F, H) correspond to median values. p Values in (B, E, G) calculated with log-rank test and in (C, F, H) with Mann-Whitney test. Peritoneal cavity (PC). *p<0.05, **p<0.01, ***p<0.001, ns: not significant.

The online version of this article includes the following figure supplement(s) for figure 4:

**Figure supplement 1.** *Ggta1* deletion reduces microbiota pathogenicity.

mutations in these genes are rare, and in some cases unique, to the human lineage. The latter is illustrated by the loss of CMP-*N*-acetylneuraminic acid hydroxylase (*CMAH*) function, which eliminated expression of the sialic acid, *N*-glycolylneuraminic acid (Neu5Gc), in humans (*Ghaderi et al., 2011*).

Co-evolution of ancestral hominids with commensal bacteria in their microbiota (*Dethlefsen et al., 2007*; *Huttenhower et al., 2012*; *Moeller et al., 2016*), is thought to have provided a series of fitness advantages including, among others, optimization of nutrient intake from diet, regulation of different aspects of organismal metabolism or colonization resistance against pathogenic bacteria (*Buffie and Pamer, 2013*). Here we propose that the loss of αGal expression, as it occurred during primate evolution, might have exerted a major impact on the nature of these symbiotic associations. In support of this notion, we found that *Ggta1* deletion in mice was associated with major changes in the composition of the gut microbiota (*Figure 1*). This occurred over several generations under experimental conditions of exposure to environmental-derived pathobionts, and minimum relative impact exerted by other environmental factors on microbiota composition (*Ubeda et al., 2012*), arguing that the *Ggta1* genotype modulates the microbiota composition. On the basis of this observation alone (*Figure 1*), one cannot exclude the observed divergence in the microbiota bacterial population frequencies harbored by wild type vs. *Ggta1*-deleted mice (*Figure 1*) from being a stochastic event. However, the observation that these changes occur via an Ig-dependent mechanism that differs in wild type vs. *Ggta1*-deleted mice (*Figure 3*) does suggest that loss of αGal contributes critically to shape the microbiota composition of *Ggta1-deleted* mice.

While our studies do not provide direct evidence that the loss of αGal played a major role in shaping the microbiota composition of Old- *vs.* New World primates, this notion is supported, indirectly, by the recent finding that mutations altering the expression of human ABO blood group glycans are associated with shaping of the bacterial composition of the gut microbiota (*Rühlemann et al., 2021*). In light of the structural similarity between the human B blood group (i. e. Galα1-3(Fucα1-2)Galβ1-4GlcNAcβ1-R), and αGal (i.e. Galα1-3Galβ1-4GlcNAcβ1-R) glycans, our findings probably reflect a general mechanism whereby host 'αGal-like' glycans shape the microbiota composition of mice and primates, including that of humans. Whether or not this is the case, considering that microbiota composition is shaped mostly by environmental factors, remains to be formally established.

The mechanism via which *Ggta1* deletion shapes the gut microbiota is associated with targeting of αGal-expressing bacteria by IgA (*Figure 2*). Consistent with this notion, a relatively large proportion of probiotic bacteria as well as bacteria present in the mouse microbiota express αGal-like glycans at the cell surface (*Figure 2*). About 30% of these carry IgA at the cell surface in the mouse microbiota and, as such, are considered as immunogenic (*Palm et al., 2014*). The proportion of IgA⁺αGal⁺ bacteria was reduced in the microbiota of *Ggta1*-deleted mice and was presumably eliminated (*Figure 2*). This suggests that *Ggta1* deletion probably broadens bacterial recognition by IgA to include immunogenic αGal⁺ bacteria, which as a result are probably purged from the microbiota. Whether this occurs via a mechanism involving the recognition of αGal-like glycans, and/or other related epitopes expressed at the surface of these bacteria, by IgA, has not been established. Of note, these are not mutually exclusive possibilities since: (i) IgA can target αGal-like glycans and modulate bacterial pathogenicity (*Hamadeh et al., 1995*), (ii) IgA are poly-reactive and can target common antigens expressed by bacteria (*Bunker et al., 2017*; *Bunker et al., 2015*) and (iii) αGal-reactive antibodies present a degree of poly-reactivity against non-αGal related bacterial epitopes (*Bernth Jensen et al., 2021*).

The identity of the αGal⁺ bacteria targeted by IgA remains elusive but likely includes Gram-negative pathobionts from the *Enterobacteriaceae* family, as demonstrated for *Escherichia* (*E.*) *coli* O86: B7 (*Yilmaz et al., 2014*), which expresses the αGal-like glycan Galα1-3Gal(Fucα1–2)β1-3GlcNAcβ1-4Glc as part of the lipopolysaccharide (LPS) O-antigen (*Guo et al., 2005*). Of note, this pathobiont can induce a systemic αGal-specific NAb response in humans (*Springer and Horton, 1969*) as well as in *Ggta1*-deleted mice, which is protective against infection by pathogens expressing αGal-like glycans (*Yilmaz et al., 2014*). The finding that several commensal bacteria in the human gut microbiota express αGal-like glycans (*Figure 2—figure supplement 2*) suggests that other bacteria might contribute to this protective response.

Our findings also suggest that *Ggta1* deletion shapes the bacterial community structure among highly abundant bacterial taxa in the microbiota via an Ig-dependent mechanism, and among low abundant bacterial taxa independently of Ig (*Figure 3*). Presumably, shaping of highly abundant taxa by Ig occurs via a mechanism that involves not only IgA produced by the offspring, but also maternal IgG transferred through the placenta as well as IgM, IgG, and IgA transferred through maternal milk (*Koch et al., 2016*). These regulate neonatal innate (*Gomez de Agüero et al., 2016*) and adaptive (*Koch et al., 2016*) immunity, shaping the offspring microbiota composition (*Gensollen et al., 2016*; *Koch et al., 2016*). Whether this occurs via targeting of αGal-like glycans, as discussed above, and/or via other bacterial antigens expressed by immunogenic bacteria has not been established.

Shaping of lowly abundant bacterial taxa independently of Ig (*Figure 3*) suggests that other mechanisms contribute to shaping the microbiota of *Ggta1*-deleted mice. These probably include the modulation of nutritional or spatial niches due to the loss of αGal from complex glycosylated structures present at epithelial barriers, such as the mucus, as demonstrated for other glycans (*Pickard et al., 2014*).

The selective pressure exerted by the adaptive immune system of *Ggta1⁻/⁻* mice on the bacterial species present in the microbiota, probably allows for the establishment of a more diverse ecosystem containing pathobionts (*Singh et al., 2021*), such as Helicobactereaceae (*Figure 1—figure supplements 1* and *2*). These can elicit the production of antibodies targeting and exerting negative selective pressure on other bacterial symbionts and releasing ecological niches, thus further shaping the microbiota. Expansion of these pathobionts in the microbiota of *Ggta1⁻/⁻* mice is restrained by Ig, which probably explains the lack of associated pathology (*Figure 1—figure supplement 3*). This suggests that the loss of *GGTA1* function in ancestral primates fostered mutualistic interactions with more diverse bacterial ecosystems, incorporating pathobionts such as *Helicobacter pylori*, associated with fitness costs (*Atherton and Blaser, 2009*) and gains (*Linz et al., 2007*).

Our findings suggest that Ig-shaping of the bacterial species present in the gut microbiota of *Ggta1⁻/⁻* mice reduces the microbiota pathogenicity, as illustrated when comparing the lethal outcome of systemic infections using the Ig-shaped *vs.* non-shaped microbiota (*Figure 4A*, *Figure 4—figure supplement 1A and B*). This reduction in pathogenicity means that the effector mechanism underlying resistance against systemic infection by the Ig-shaped microbiota no longer relies on the adaptive immune system (*Figure 4—figure supplement 1C and E*), but instead on cellular components of the innate immune systems, namely, neutrophils (*Figure 4G,H*).

We propose that *Ggta1* deletion in mice increases resistance to bacterial sepsis via two distinct antibody-dependent mechanisms. The first involves a relative increase in antibody effector function, when αGal is not present in the biantennary glycan structures of IgG (*Singh et al., 2021*). The second relies on shaping and reduction of the microbiota pathogenicity by antibodies, most likely from the IgA isotype. To what extent the lack of αGal from the biantennary glycan structures of Ig also contributes to shape and reduce the microbiota pathogenicity, remains to be established. It is possible however, that similar to IgG (*Singh et al., 2021*), the absence of αGal from the glycan structures of different Ig isotypes, including IgA, modulates their effector function, when targeting immunogenic bacteria in the microbiota.

The notion of host mechanisms shaping the microbiota composition toward a reduction of its pathogenicity is in keeping with host microbial interactions not being hardwired, but instead shifting between symbiotic to pathogenic depending on host and microbe cooperative behaviors (*Ayres, 2016*; *Vonaesch et al., 2018*). For example, when restricted to the microbiota, bacterial pathobionts can behave as commensals, posing no pathogenic threat to the host (*Vonaesch et al., 2018*), while triggering sepsis (*Haak and Wiersinga, 2017*) upon translocation across epithelial barriers (*Caruso et al., 2020*). The high fitness cost imposed to modern humans by sepsis (*Rudd et al.,*

*2020*) suggests that mutations shaping the composition of the microbiota toward a reduced capacity to elicit sepsis should be associated with major fitness advantages. Our findings suggest that loss-of-function mutations in *GGTA1* act in such a manner.

When challenged experimentally with a bacterial inoculum or bacterial lipopolysaccharide (LPS), non-human primates appear to be far more susceptible to develop sepsis or septic shock, respectively, as compared to other mammalian lineages (*Chen et al., 2019*). This may appear at odds with our interpretation that similar to *Ggta1* deletion in mice (*Figure 4*; *Singh et al., 2021*), the evolutionary loss of *GGTA1* function in Old-World primates might have increased resistance to bacterial sepsis (*Singh et al., 2021*). One likely explanation for this apparent discrepancy may relate to the intrinsic properties of primate immunity, whereby natural selection of traits enhancing immune-driven resistance mechanisms (*Olson, 1999*; *Wang et al., 2006*) might be associated with lower capacity to establish disease tolerance to infection (*Martins et al., 2019*; *Medzhitov et al., 2012*). This interpretation is consistent with resistance and disease tolerance to infection being negatively correlated (*Råberg et al., 2007*), such that traits increasing resistance might be associated with a reduction in disease tolerance, as an evolutionary trade-off.

In conclusion, protective immunity against αGal-expressing pathogens was likely a major driving force in the natural selective events that led to the fixation of loss-of-function mutations in the *GGTA1* gene of ancestral Old World primates (*Galili, 2016*; *Soares and Yilmaz, 2016*). Moreover, in the absence of αGal, the glycan structures associated with the Fc portion of IgG, can increase IgG-effector function and resistance to bacterial infections, irrespectively of αGal-specific immunity (*Singh et al., 2021*). The net survival advantage against infection provided by these traits came alongside the emergence of reproductive senescence (*Singh et al., 2021*), a major fitness cost presumably outweighed by endemic exposure to highly virulent pathogens (*Singh et al., 2021*). Here we provide experimental evidence for yet another fitness advantage against infection, associated with the loss of *GGTA1*, driven by Ig-shaping and reduction of microbiota pathogenicity. We infer that ancestral Old World primates carrying loss-of-function mutations in *GGTA1* probably shaped their microbiota to minimize its pathogenic effect, providing a major fitness advantage against sepsis.

## Materials and methods

### Mice

Mice were used in accordance with protocols approved by the Ethics Committee of the Instituto Gulbenkian de Ciência (IGC) and Direção Geral de Alimentação e Veterinária (DGAV), following the Portuguese (Decreto-Lei no. 113/2013) and European (directive 2010/63/EU) legislation for animal housing, husbandry, and welfare. C57BL/6J wild-type (*Ggta1$^{+/+}$*), *Ggta1$^{-/-}$* (*Tearle et al., 1996*), *Igh-J$^{-/-}$Ggta1$^{-/-}$* (*Gu et al., 1993*), *Tcrb$^{-/-}$Ggta1$^{-/-}$* (*Yilmaz et al., 2014*), *Igha$^{-/-}$Ggta1$^{-/-}$* (*Singh et al., 2021*), and *Rag2$^{-/-}$Ggta1$^{-/-}$* (*Singh et al., 2021*) mice were used. Mice were bred and maintained under specific pathogen-free (SPF) conditions (12 hr day/night, fed ad libitum), as described (*Yilmaz et al., 2014*). Germ-free (GF) C57BL/6J *Ggta1$^{+/+}$* and *Ggta1$^{-/-}$* animals were bred and raised in the IGC gnotobiology facility in axenic isolators (La Calhene/ORM), as described (*Yilmaz et al., 2014*; *Singh et al., 2021*). Sterility of food, water, bedding, oral swab, and feces were confirmed as described (*Singh et al., 2021*). Both male and female mice were used for all experiments. All animals were studied between 9–16 weeks of age unless otherwise indicated.

### Breeding experiments

Vertical transmission of the microbiota from *Ggta1$^{+/+}$* mice to *Ggta1$^{-/-}$* and *Ggta1$^{+/+}$* offspring over several generations was achieved, as described (*Ubeda et al., 2012*; *Singh et al., 2021*). Briefly, two or more breeding groups were established, consisting of two *Ggta1$^{+/+}$* females and one *Ggta1$^{-/-}$* male per cage. The male was removed after one week and the females were placed in a clean cage until delivery. F$_1$ *Ggta1$^{+/-}$* mice were weaned at 3–4 weeks of age and then co-housed until 8 weeks of age. Two or more F$_1$ *Ggta1$^{+/-}$* groups were established randomly using one male and two females per cage. F$_2$ pups were weaned at 3–4 weeks of age, genotyped and segregated according to their *Ggta1$^{-/-}$* vs. *Ggta1$^{+/+}$* genotype in separate cages until adulthood. F$_3$ to F$_5$ pups were generated in a

similar manner. Fecal pellets from two to three cages per genotype were collected (10–12 weeks of age) for microbiota analysis.

The effect of *Ggta1* genotype and Ig on microbiota composition derived from *Ggta1$^{+/+}$* mice was achieved essentially as described (*Singh et al., 2021*). Briefly, two or more breeding groups were established, consisting of two *Igh-J$^{+/+}$Ggta1$^{+/+}$* females and one *Igh-J$^{-/-}$Ggta1$^{-/-}$* male per cage. The male was removed after 1 week and the females were placed in a clean cage until delivery. F$_1$ *Igh-J$^{+/-}$Ggta1$^{+/-}$* pups were kept with mothers until weaning at 3–4 weeks of age and co-housed until 8 weeks of age. Two or more F$_1$ *Igh-J$^{+/-}$Ggta1$^{+/-}$* breeding groups were established randomly using one male and two females per cage. Littermate F$_2$ pups were weaned at 3–4 weeks of age, genotyped and segregated according to their *Igh-J$^{+/+}$Ggta1$^{-/-}$*, *Igh-J$^{-/-}$Ggta1$^{-/-}$*, *Igh-J$^{+/+}$Ggta1$^{+/+}$* and *Igh-J$^{-/-}$Ggta1$^{+/+}$* genotypes in separate cages until adulthood. F$_3$ pups were generated in a similar manner. Fecal pellets from two to three cages per genotype were collected (10–12 weeks of age) for microbiota analysis.

## Genotyping

Mice were genotyped from tail biopsies (0.5–1 cm) by PCR of genomic DNA as per manufacturer's protocols (KAPA mouse genotyping kit #KK7352) as described (*Singh et al., 2021*).

## Cecal slurry injection

Cecal slurry injection was performed essentially as described (*Singh et al., 2021*). Microbiota pathogenicity experiments were performed by preparing cecal slurry from *Ggta1$^{-/-}$* vs. *Rag2$^{-/-}$Ggta1$^{-/-}$* mice or from *Ggta1$^{+/+}$* vs. *Rag2$^{-/-}$Ggta1$^{+/+}$* mice and injecting to recipient mice (*i.p.* 1 mg/g body weight) in parallel. Mice were monitored every 12 hr for survival for 14 days or euthanized at various time points for analysis of different parameters.

## Pathogen load

Quantification of pathogen load in the mice was performed 24 hr after cecal slurry injection, essentially as described (*Singh et al., 2021*).

## Neutrophil depletion

Anti-Gr1 mAb (Clone: RB6-8C5, 300 µg in 200 µL PBS) was injected (i.p.) to mice 24 hr before cecal slurry injection. Neutrophil depletion was confirmed by flow cytometry, using CD11b$^+$Ly6G$^+$ cell staining in the blood.

## Monocyte/macrophage depletion

Clodronate or PBS liposomes (http://www.Clodronateliposomes.org) was injected (10 µL/g, i.p.) to mice 72 hr before cecal slurry injection. Monocyte/macrophage depletion was confirmed by flow cytometry, using CD11b$^+$F4/80$^+$ and CD11b$^+$Ly6C$^+$ staining in the peritoneal lavage.

## ELISA

ELISA for IgA binding to cecal extracts was done essentially as described (*Kamada et al., 2015*; *Zeng et al., 2016*). Cecal lysate was prepared as described (*Singh et al., 2021*). 96-well ELISA plates (Nunc MaxiSorp #442404) were coated with the cecal lysate (100 µL/well, 4°C, overnight), washed (3x, PBS 0.05% Tween-20, Sigma-Aldrich #P7949-500ML), blocked (200 µL, PBS 1% BSA wt/vol, Calbiochem #12659–100 GM, 3 hr, RT) and washed (3x, PBS 0.05% Tween-20). Plates were incubated with serially diluted (50 µL) mouse sera (1:20 to 1:100 in PBS 1% BSA, wt/vol, 2 hr, RT) and washed (5x, PBS/0.05% Tween-20). IgA was detected using horseradish peroxidase (HRP)-conjugated goat anti-mouse IgA (Southern Biotech #1040–05), in PBS/1%BSA/0.01% Tween-20 (100 µL, 1:4000 vol/vol, 1 hr, RT) and plates were washed (5x, PBS/0.05% Tween-20).

For quantification of total serum and small intestinal IgA, 96-well ELISA plates (Nunc MaxiSorp #442404) were coated with goat anti-mouse IgA (Southern Biotech #1040–01, 2 µg/mL in Carbonate-Bicarbonate buffer, 100 µL/well, overnight, 4°C), washed (3x, PBS 0.05% Tween-20, Sigma-Aldrich #P7949-500ML), blocked (200 µL, PBS 1% BSA wt/vol, 2 hr, RT) and washed (3x, PBS 0.05% Tween-20). Plates were incubated with serially diluted serum or gut content (50 µL, PBS 1% BSA, wt/vol, 2 hr, RT) and standard mouse IgA (Southern Biotech #0106–01, prepared in duplicates) and

washed (5x, PBS/0.05% Tween-20). IgA was detected using HRP-conjugated goat anti-mouse IgA (Southern Biotech #1040–05) in PBS/1%BSA/0.01% Tween-20 (100 µL, 1:4000 vol/vol, 1 hr, RT) and plates were washed (5x, PBS/0.05% Tween-20).

For quantification of anti-αGal IgG, IgM and IgA, 96-well ELISA plates (Nunc MaxiSorp #442404) were coated with αGal-BSA (Dextra, 5 µg/mL in Carbonate-Bicarbonate buffer, 50 µL/well, overnight, 4°C). Wells were washed (3x, PBS 0.05% Tween-20, Sigma-Aldrich #P7949-500ML), blocked (200 µL, PBS 1% BSA wt/vol, 2 hr, RT) and washed (3x, PBS 0.05% Tween-20). Plates were incubated with serially diluted serum or fecal content (50 µL, PBS 1% BSA, wt/vol, 2 hr, RT) and standard mouse anti-αGal IgG, IgM or IgA and washed (5x, PBS/0.05% Tween-20). Anti-αGal Abs were detected using HRP-conjugated goat anti-mouse IgG, IgM, and IgA in PBS/1%BSA (100 µL, 1:4000 vol/vol, 1.5 hr, RT) and plates were washed (5x, PBS/0.05% Tween-20).

HRP activity was detected with 3,3′,5,5′-Tetramethylbenzidine (TMB) Substrate Reagent (BD Biosciences #555214, 50 µL, 20–25 min, dark, RT) and the reaction was stopped using sulfuric acid (2N, 50 µL). Optical densities (OD) were measured using a MultiSkan Go spectrophotometer (Thermo-Fisher) at $\lambda$ = 450 nm and normalized by subtracting background OD values ($\lambda$ = 600 nm).

For measurement of fecal Lipocalin (Lcn-2), feces were resuspended with sterile PBS (100 mg/mL), vortexed (5 min.), and centrifuged (12,000 rpm, 4°C, 10 min). Lcn-2 levels were determined in fecal supernatants using a LEGEND MAX Mouse NGAL (Lipocalin-2) ELISA Kit (BioLegend #443708).

## Flow cytometry of bacterial staining for IgA and αGal

Overnight cultures of bacteria were prepared as described in the next section. Samples of 5–20 µL of each bacterial culture depending on $OD_{600}$ measurements, and corresponding to approximately $10^6$–$10^7$ cells, were fixed in paraformaldehyde (PFA; 4% wt/vol in PBS) and washed with filter-sterilized PBS. For detection of IgA binding and αGal expression by gut microbes, small intestinal, cecal, colon and fecal samples were homogenized (5 mg/ml in PBS) by vortexing (maximum speed, 5 min, RT) and filtered (BD Falcon, 40 µm cell strainer # 352340). Larger debris were pelleted by centrifugation (600 g, 4°C, 5 min). Fifty µL supernatant (containing bacteria) per condition was added to a 96-well v-bottom plate (Corning Costar #3894) for staining. Bacteria were pelleted by centrifugation (3700 g, 10 min, 4°C) and suspended in flow cytometry buffer (filter-sterilized 1xPBS, 2% BSA, wt/vol). Bacterial DNA was stained using SYTO41 Blue Fluorescent Nucleic Acid Stain (Molecular Probes #S11352, 1:200 vol/vol, wt/vol) in flow cytometry buffer (100 µL, 30 min, RT). Cecal content from germ-free (GF) mice was used as control. Bacteria were washed in flow cytometry buffer (200 µL), centrifuged (4000 g, 10 min, 4°C) and supernatant was removed by flicking the plate. Bacteria were incubated in Fluorescein Isothiocyanate (FITC)-conjugated BSI-B4 lectin from *Bandeiraea* (*Griffonia*) *simplicifolia* (Sigma-Aldrich, #L2895-1MG, 50 µL, 40 µg/mL in PBS, 2 hr) for detection of αGal and anti-mouse IgA-PE mAb (mA-6E1, eBiosciences # 12-4204-82, 1:100 in PBS 2% BSA wt/vol, 30 min) and washed as above. *E coli* O86:B7 (about $10^7$ per tube) was used as a positive control for bacterial αGal expression. Samples were re-suspended in flow cytometry buffer (300 µL), transferred to round-bottom tubes (BD Falcon #352235) and centrifuged (300 g, 1 min., RT). Samples were analyzed in LSR Fortessa SORP (BD Biosciences) equipped with a high-throughput sampler (HTS) using the FACSDiva Software (BD v.6.2) and analyzed by FlowJo software (v10.0.7) as described (*Singh et al., 2021*).

## Detection of αGal expression by probiotic bacteria

Lyophilised stocks of probiotic strains of *Bifidobacterium bifidum* BB-G90, *B. breve* BBG95, *B. infantis* BIG201, *B. longum* BLG301, *B. lactis* BLG101, *Lactobacillus bulgaricus* LBG40, *L. casei* LCG11, *L. cremoris* LLCG42, *L. lactis* LLLG25, *L. paracasei* LPcG110, L. plantarum LPG18, *L. reuteri* LrG100, *L. rhamnosus* LrG14, *L. salivarius* LSG60, *L. acidophilus* LAG80, *Streptococcus thermophilus* STG30, *L. helveticus* LHG444, *B. infantis* BIG191, *L. acidophilus* LAG436, *L. bulgaricus* LBG058, *L. bulgaricus* LBG117, *L. casei* LCG402, *L. plantarum* LPG425, *L. plantarum* LPG429, *L. plantarum* LPG443, *L. rhamnosus* LrG426 (BioGrowing; http://www.biogrowing.com/html/en/index.php?ac=article&at=list&tid=122), were resuspended in PBS (2 g/10 mL). Ten-fold serial dilutions in PBS were made and 100 µL plated onto MRS agar plates. Undiluted stock solutions were also streaked onto MRS agar plates. Bacteria were grown under anaerobic conditions using the GasPak EZ Anaerobe system (37°C, overnight). Cells from several independent colonies were selected and diluted in 200 µL PBS,

fixed with PFA (50 µL, 16% solution, final concentration: 3.2% PFA, 30 min, 21°C), before being washed in PBS. Cells were stained in BSI-B$_4$-FITC lectin (*Bandeiraea* (*Griffonia*) *simplicifolia* Isolectin B$_4$ (BSI-B$_4$), FITC conjugate, L2895 SIGMA) (40 mg/mL, 45 min, 21°C, dark), before being washed in PBS and analyzed by flow cytometry using an LSR Fortessa SORP (BD), as described above. As positive and negative controls, *Escherichia coli* O86:B7 (ATCC 12701; Rockville, Md.), and *E. coli* K12 (ATCC 10798), respectively, were cultured in liquid Luria Bertani medium (37°C, overnight, aeration), washed twice (PBS; 4000 g; 5 min; 4°C) and re-suspended in PBS. Bacteria were fixed (200 µL; 4% PFA in PBS; 20–30 min; RT) and washed (1X; PBS; 4000 g; 5 min; RT). $10^8$–$10^9$ CFU/mL were stained with BSI-B$_4$-FITC (200 µL; 40 µg/mL; 2 hr, RT) for flow cytometric analysis also as described above.

## Extraction of bacterial DNA from feces
Bacterial DNA was extracted from fecal pellets (QIAamp Fast DNA Stool Mini Kit #50951604) as described (*Singh et al., 2021*).

## Amplicon sequencing: 16S amplicons sequencing and analysis
The 16S rRNA V4 region was amplified and sequenced following the Earth Microbiome Project (http://www.earthmicrobiome.org/emp-standard-protocols/), and analyzed using QIIME 1.9.1 as described (*Singh et al., 2021*). 16S sequencing data was submitted to the Sequence Read Archive (SRA), with the BioProject reference PRJNA701192 (accession code).

## Statistical analysis
Statistical tests were performed using GraphPad Prism Software (v.6.0). All statistical details, including statistical tests, exact value of n, what n represents, definition of center, dispersion and precision measures are provided in each figure legend.

# Acknowledgements
We thank our colleagues J Howard and I Gordo (Instituto Gulbenkian de Ciência; IGC) for critical review of the manuscript, IGC Genomics, Flow Cytometry, Antibody, Histopathology and Animal House Facilities. SS was supported by Fundação para a Ciência e Tecnologia (FCT; SFRH/BD/52177/2013), JAT by an ESCMID Research Grant and FCT (SFRH/BPD/112135/2015) and MPS by the Gulbenkian, 'La Caixa' (HR18-00502) and Bill and Melinda Gates (OPP1148170) Foundations, FCT (5723/2014 and FEDER029411). MPS is an associate member of the Deutsche Forschungsgemeinschaft (DFG, German Research Foundation) Cluster of Excellence 'Balance of the Microverse' (https://microverse-cluster.de/en).

# Additional information

## Funding

| Funder | Grant reference number | Author |
| --- | --- | --- |
| Fundação para a Ciência e a Tecnologia | SFRH/BD/52177/2013 | Sumnima Singh |
| Fundação para a Ciência e a Tecnologia | SFRH/BPD/112135/2015 | Miguel Soares |
| European Society of Clinical Microbiology and Infectious Diseases | | Jessica Ann Thompson |
| Fundação Calouste Gulbenkian | 5723/2014 and FEDER029411 | Miguel Soares |
| Bill and Melinda Gates Foundation | OPP1148170 | Miguel Soares |
| "la Caixa" Foundation | HR18-00502 | Miguel Soares |

The funders had no role in study design, data collection and interpretation, or the decision to submit the work for publication.

## Author contributions
Sumnima Singh, Miguel P Soares, Conceptualization, Data curation, Formal analysis, Supervision, Funding acquisition, Investigation, Methodology, Writing - original draft, Project administration, Writing - review and editing; Patricia Bastos-Amador, Conceptualization, Data curation, Formal analysis, Investigation, Methodology, Writing - original draft, Writing - review and editing; Jessica Ann Thompson, Formal analysis, Funding acquisition, Investigation, Methodology, Writing - original draft, Writing - review and editing; Mauro Truglio, Data curation, Formal analysis; Bahtiyar Yilmaz, Data curation, Formal analysis, Investigation, Writing - review and editing; Silvia Cardoso, Formal analysis, Investigation, Methodology, Writing - review and editing; Daniel Sobral, Data curation, Software, Formal analysis, Investigation, Methodology, Writing - review and editing

## Author ORCIDs
Sumnima Singh  https://orcid.org/0000-0002-1826-0180
Daniel Sobral  https://orcid.org/0000-0003-3955-0117
Miguel P Soares  https://orcid.org/0000-0002-9314-4833

## Ethics
Animal experimentation: Mice were used in accordance with protocols approved by the Ethics Committee of the Instituto Gulbenkian de Ciência (IGC) and Direção Geral de Alimentação e Veterinária (DGAV), following the Portuguese (Decreto-Lei no. 113/2013) and European (directive 2010/63/EU) legislation for animal housing, husbandry and welfare.

## Decision letter and Author response
Decision letter https://doi.org/10.7554/eLife.67450.sa1
Author response https://doi.org/10.7554/eLife.67450.sa2

# Additional files
## Supplementary files
• Transparent reporting form

## Data availability
All data generated during this study are included in the manuscript and supporting files. The 16S sequencing data can be accessed via NCBI Bioproject PRJNA701192.

The following dataset was generated:

| Author(s) | Year | Dataset title | Dataset URL | Database and Identifier |
|---|---|---|---|---|
| Soares M, Singh S, Bastos-Amador P, Thompson JA, Yilmaz B, Cardoso S, Sobral D | 2021 | GLYCAN-BASED SHAPING OF THE MICROBIOTA DURING PRIMATE EVOLUTION | https://www.ncbi.nlm.nih.gov/bioproject/PRJNA701192 | NCBI BioProject, PRJNA701192 |

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

# Appendix 1

**Appendix 1—key resources table**

| Reagent type (species) or resource | Designation | Source or reference | Identifiers | Additional information |
|---|---|---|---|---|
| Strain, strain background (*Mus musculus*) | B6.C57BL/6/J: *Ggta1*$^{+/+}$ | Obtained originally from Jackson Laboratory and maintained at IGC | N/A | |
| Strain, strain background (*Mus musculus*) | B6. *Ggta1*$^{-/-}$ | Obtained originally from *Tearle et al., 1996* and maintained at IGC | N/A | |
| Strain, strain background (*Mus musculus*) | B6. *Igha*$^{-/-}$*Ggta1*$^{-/-}$ | *Singh et al., 2021* | N/A | |
| Strain, strain background (*Mus musculus*) | B6. *Igh-J*$^{-/-}$*Ggta1*$^{-/-}$ | *Yilmaz et al., 2014* | N/A | |
| Strain, strain background (*Mus musculus*) | B6. *Rag2*$^{-/-}$*Ggta1*$^{-/-}$ | *Singh et al., 2021* | N/A | |
| Strain, strain background (*Mus musculus*) | B6. *Tcrb*$^{-/-}$*Ggta1*$^{-/-}$ | *Yilmaz et al., 2014* | N/A | |
| Strain, strain background (*Bifidobacterium bifidum*) | *Bifidobacterium bifidum* BB-G90 | BioGrowing Probiotics | N/A | See Materials and methods, Section Detection of αGal expression by probiotic bacteria |
| Strain, strain background (*Bifidobacterium breve*) | *Bifidobacterium breve* BB-G95 | BioGrowing Probiotics | N/A | See Materials and methods, Section Detection of αGal expression by probiotic bacteria |
| Strain, strain background (*Bifidobacterium infantis*) | *Bifidobacterium infantis* BI-G191 | BioGrowing Probiotics | N/A | See Materials and methods, Section Detection of αGal expression by probiotic bacteria |
| Strain, strain background (*Bifidobacterium infantis*) | *Bifidobacterium infantis* BI-G201 | BioGrowing Probiotics | N/A | See Materials and methods, Section Detection of αGal expression by probiotic bacteria |
| Strain, strain background (*Bifidobacterium lactis*) | *Bifidobacterium lactis* BL-G101 | BioGrowing Probiotics | N/A | See Materials and methods, Section Detection of αGal expression by probiotic bacteria |
| Strain, strain background (*Bifidobacterium longum*) | *Bifidobacterium longum* BL-G301 | BioGrowing Probiotics | N/A | See Materials and methods, Section Detection of αGal expression by probiotic bacteria |

*Continued on next page*

*Appendix 1—key resources table continued*

| Reagent type (species) or resource | Designation | Source or reference | Identifiers | Additional information |
|---|---|---|---|---|
| Strain, strain background (*Escherichia coli*) | *Escherichia coli* K12 | ATCC | ATCC10798 | See Materials and methods, Section Detection of αGal expression by probiotic bacteria |
| Strain, strain background (*Escherichia coli*) | *Escherichia coli* O86:B7 | ATCC | ATCC12701 | See Materials and methods, Section Detection of αGal expression by probiotic bacteria |
| Strain, strain background (*Lactobacillus acidophilus*) | *Lactobacillus acidophilus* LA-G80 | BioGrowing Probiotics | N/A | See Materials and methods, Section Detection of αGal expression by probiotic bacteria |
| Strain, strain background (*Lactobacillus acidophilus*) | *Lactobacillus acidophilus* LA-G436 | BioGrowing Probiotics | N/A | See Materials and methods, Section Detection of αGal expression by probiotic bacteria |
| Strain, strain background (*Lactobacillus bulgaricus*) | *Lactobacillus bulgaricus* LB-G4058 | BioGrowing Probiotics | N/A | See Materials and methods, Section Detection of αGal expression by probiotic bacteria |
| Strain, strain background (*Lactobacillus bulgaricus*) | *Lactobacillus bulgaricus* LB-G117 | BioGrowing Probiotics | N/A | See Materials and methods, Section Detection of αGal expression by probiotic bacteria |
| Strain, strain background (*Lactobacillus bulgaricus*) | *Lactobacillus bulgaricus* LB-G40 | BioGrowing Probiotics | N/A | See Materials and methods, Section Detection of αGal expression by probiotic bacteria |
| Strain, strain background (*Lactobacillus casei*) | *Lactobacillus casei* LC-G11 | BioGrowing Probiotics | N/A | See Materials and methods, Section Detection of αGal expression by probiotic bacteria |
| Strain, strain background (*Lactobacillus cremoris*) | *Lactobacillus cremoris* LLC-G42 | BioGrowing Probiotics | N/A | See Materials and methods, Section Detection of αGal expression by probiotic bacteria |
| Strain, strain background (*Lactobacillus helveticus*) | *Lactobacillus helveticus* LH-G444 | BioGrowing Probiotics | N/A | See Materials and methods, Section Detection of αGal expression by probiotic bacteria |
| Strain, strain background (*Lactobacillus casei*) | *Lactobacillus casei* LC-G402 | BioGrowing Probiotics | N/A | See Materials and methods, Section Detection of αGal expression by probiotic bacteria |

*Continued on next page*

*Appendix 1—key resources table continued*

| Reagent type (species) or resource | Designation | Source or reference | Identifiers | Additional information |
|---|---|---|---|---|
| Strain, strain background (*Lactobacillus lactis*) | *Lactobacillus lactis* LLL-G25 | BioGrowing Probiotics | N/A | See Materials and methods, Section Detection of αGal expression by probiotic bacteria |
| Strain, strain background (*Lactobacillus plantarum*) | *Lactobacillus plantarum* Lp-G18 | BioGrowing Probiotics | N/A | See Materials and methods, Section Detection of αGal expression by probiotic bacteria |
| Strain, strain background (*Lactobacillus paracasei*) | *Lactobacillus paracasei* LpC-G110 | BioGrowing Probiotics | N/A | See Materials and methods, Section Detection of αGal expression by probiotic bacteria |
| Strain, strain background (*Lactobacillus plantarum*) | *Lactobacillus plantarum* Lp-G425 | BioGrowing Probiotics | N/A | See Materials and methods, Section Detection of αGal expression by probiotic bacteria |
| Strain, strain background (*Lactobacillus plantarum*) | *Lactobacillus plantarum* Lp-G429 | BioGrowing Probiotics | N/A | See Materials and methods, Section Detection of αGal expression by probiotic bacteria |
| Strain, strain background (*Lactobacillus plantarum*) | *Lactobacillus plantarum* Lp-G443 | BioGrowing Probiotics | N/A | See Materials and methods, Section Detection of αGal expression by probiotic bacteria |
| Strain, strain background (*Lactobacillus reuteri*) | *Lactobacillus reuteri* Lr-G100 | BioGrowing Probiotics | N/A | See Materials and methods, Section Detection of αGal expression by probiotic bacteria |
| Strain, strain background (*Lactobacillus rhamnosus*) | *Lactobacillus rhamnosus* Lr-G14 | BioGrowing Probiotics | N/A | See Materials and methods, Section Detection of αGal expression by probiotic bacteria |
| Strain, strain background (*Lactobacillus rhamnosus*) | *Lactobacillus rhamnosus* Lr-G26 | BioGrowing Probiotics | N/A | See Materials and methods, Section Detection of αGal expression by probiotic bacteria |
| Strain, strain background (*Lactobacillus salivarus*) | *Lactobacillus salivarus* LS-G60 | BioGrowing Probiotics | N/A | See Materials and methods, Section Detection of αGal expression by probiotic bacteria |
| Strain, strain background (*Streptococcus thermophilus*) | *Streptococcus thermophilus* ST-G30 | BioGrowing Probiotics | N/A | See Materials and methods, Section Detection of αGal expression by probiotic bacteria |
| Antibody | Anti-Mouse Monoclonal Gr1 (Clone: RB6-8C5) | Bio-Rad | N/A | 300 µg in 200 µL PBS per mouse |

*Continued on next page*

*Appendix 1—key resources table continued*

| Reagent type (species) or resource | Designation | Source or reference | Identifiers | Additional information |
|---|---|---|---|---|
| Antibody | Anti-Mouse Monoclonal IgA (mA-6E1) | eBioscience, Thermo Fisher Scientific | Cat# 12-4204-82, RRID:AB_465917 | 1:100 (Flow cytometry) |
| Antibody | Goat Anti-Mouse Polyclonal IgA-HRP | Southern Biotech | Cat# 1040–05; RRID:AB_2714213 | 1:4000 (ELISA) |
| Antibody | Goat Anti-Mouse Polyclonal IgA-Unlabelled | Southern Biotech | Cat# 1040–01, RRID:AB_2314669 | 2 µg/mL (ELISA) |
| Antibody | Goat Anti-Mouse Polyclonal IgG, Human ads-HRP | Southern Biotech | Cat# 1030–05, RRID:AB_2619742 | 1:4000 (ELISA) |
| Antibody | Goat Anti-Mouse Polyclonal IgM, Human ads-HRP | Southern Biotech | Cat# 1020–05, RRID:AB_2794201 | 1:4000 (ELISA) |
| Antibody | Anti-α-gal Mouse Monoclonal IgG1 | *Ding et al., 2008*; *Yilmaz et al., 2014* | N/A | Standard (ELISA). 0,5 µg/mL, 1:2 serial dilutions, 50 µL |
| Antibody | Anti-α-gal Mouse Monoclonal IgG2a | *Yilmaz et al., 2014* | N/A | Standard (ELISA). 0,5 Originally o/mL, 1:3 serial dilutions, 50 µL |
| Antibody | Anti-α-gal Mouse Monoclonal IgG2b | *Ding et al., 2008*; *Yilmaz et al., 2014* | N/A | Standard (ELISA). 0,5 µg/mL, 1:3 serial dilutions, 50 µL |
| Antibody | Anti-α-gal Mouse Monoclonal IgG3 | *Ding et al., 2008*; *Yilmaz et al., 2014* | N/A | Standard (ELISA). 0,5 µg/mL, 1:2 serial dilution |
| Antibody | Anti-α-gal Mouse Monoclonal IgM | *Yilmaz et al., 2014* | N/A | Standard (ELISA). 2 µg/mL, 1:2 serial dilutions, 50 µL |
| Antibody | Mouse IgA-Unlabelled | Southern Biotech | Cat# 0106–01, RRID:AB_2714214 | Standard 0.5 µg/mL (ELISA) |
| Commercial assay or kit | KAPA Mouse Genotyping Kit | KAPA Biosystems | Cat# KK7352 | |
| Commercial assay or kit | LEGEND MAX Mouse NGAL (Lipocalin-2) ELISA Kit | Biolegend | Cat# 443707 | |
| Commercial assay or kit | QIAamp Fast DNA Stool Mini Kit | Qiagen | Cat# 50951604 | |
| Chemical compound, drμg | Clodronate liposomes and control liposomes (PBS) | https://clodronateliposomes.com/ | SKU: CP-005–005 | 10 µL/g (in vivo depletion) |
| Chemical compound, drμg | Galα1-3Galβ1-4GlcNAc-BSA (14 atom spacer) | Dextra | Cat# NGP1334 | 5 µg/mL (ELISA) |
| Chemical compound, drμg | Lectin from Bandeiraea simplicifolia (Griffonia simplicifolia) Isolectin B4 (BSI-B4), FITC conjμgate, lyophilized powder | Sigma-Aldrich | #L2895-1MG | 40 µg/mL (Flow cytometry) |
| Chemical compound, drμg | SYTO41 Blue Fluorescent Nucleic Acid Stain | Thermofisher Scientific | Cat# S11352 | 1:200 (Flow cytometry) |

*Continued on next page*

*Appendix 1—key resources table continued*

| Reagent type (species) or resource | Designation | Source or reference | Identifiers | Additional information |
|---|---|---|---|---|
| Software, algorithm | Greengenes | *DeSantis et al., 2006* | http://greengenes.secondgenome.com/v0.13.8 | |
| Software, algorithm | QIIME | *Caporaso et al., 2010* | http://qiime.org/v0.1.9.1 | |

