## [Decision Letter]

**Acceptance summary:**

30 million years ago the ancestors of Old World primates lost the ability to produce α-gal due to the fixation of several loss-of-function mutations in the *GGTA1* gene. The evolutionary advantage of such a loss remains elusive. Here, the authors provide additional insights into the pleiotropic role of *Ggta1* in shaping the gut microbiota, immune function, susceptibility to sepsis, and eventual fitness advantage.

**Decision letter after peer review:**

Thank you for submitting your article "Glycan-Based shaping of the Microbiota during Primate Evolution" for consideration by *eLife*. Your article has been reviewed by 3 peer reviewers, and the evaluation has been overseen by a Reviewing Editor and George Perry as the Senior Editor. The following individuals involved in review of your submission have agreed to reveal their identity: Pascale Gagneux (Reviewer #1); Ruslan Medzhitov (Reviewer #2); Luis B Barreiro (Reviewer #3).

Essential revisions:

1. Please see the recommendations for authors for additional points of clarity. You're welcome to include additional data but textual edits are probably sufficient.

2. Please also address the following additional points: How do you reconcile the fact that despite not expressing a-gal, apes and OWM are much more susceptible to sepsis as compared to species that do express a-gal, such as mice? In the context of increased resistance to sepsis what seems to be more important – the remodeling of the microbiome by IgA or the increased effector function of IgG? The fact that the microbiome of laboratory mice and primates is likely very different and therefore further experiments are needed to evaluate if the findings described in this work directly translate to a mechanisms of increased resistant to sepsis in primates. Please discuss the potential ramifications of IgG glycosylation changes and after α Gal loss. Given the demonstrated role of sialylation for effector function of IgG, could it be that lack of α Gal actually leads to higher levels of terminal sialic acid on IcG Fc N-glycans, and what would be the functional effects of that?

*Reviewer #1 (Recommendations for the authors):*

CMAH/ Cmah loss-of function occurred at least 8 times concergently across mammals: Human, New-World monkey, Mustelids, twice in microbats, white tailed deer, sperm whale, and hedgehogs (Springer et al. 2014, PMID: 25124893, Ng et al. 2014 PMID: 25517696, Peri et al. 2018 PMID:29206915).

In stark contrast to α Gal, no microbe has yet been documented to be decorated with intrinsic N-glycolyl neuramninic acid, so microbiome immunization is very unlikely in that contsetx, although some human pathobionts, e.g. Hemophilus may accumulate it from the human host, where dietary Neu5Gc can become incorporated in human glycoconjugates.

The use of Bandeiraea simplicifolia plant lectin (BSL-B4) for staining of α Gal glycan epitopes is well established. Nevertheless, using a negative control, such as treatment with coffee bean galactosidase specifically acting on terminal α 1,3 Galactose to confirm the α-Gal epitopes on various microbes might be very useful.

The authors propose that the lack of GGTA in humans and other primates of Africa and Asia might have helped mitigate sepsis pathology, but they should address the well-known observation that humans are much more prone to sepsis than other mammals, *including* Old World monkeys and apes.

Many unusual things have happened to immunity on the lineage leading to modern humans. Could the authors discuss any evidence that other Old World Primates are somehow less prone to sepsis than α Gal positive mammas, including their New World primate cousins?

Page 2 Line 72 IgA limits, not limit.

*Reviewer #2 (Recommendations for the authors):*

I think the authors did a great job at investigating the consequences of GGTA deletion and its impact on host -microbiome interactions. I think the study can be accepted pretty much 'as is'.

*Reviewer #3 (Recommendations for the authors):*

One major open question is how much of the phenotypes described herein are representative of what happens in primates once they lost GGTA1 (which is ultimately what motivated the work ). Although I appreciate that similar experiments cannot be performed in primates there are a few experiments that would be doable. For example, the authors could analyze microbiome data from New- vs Old-world primates (humans included) to see if the former (who still express a-gal) do indeed have a less pathogenic microbiome (there is data out there already available). Another experiment would be to perform the sepsis challenges using cecal inoculum derived from New- vs Old-world primates/humans and see if the microbiome from New world monkeys is indeed less pathogenic than that from Old-world primates, as predicted by their model. In the absence of doing such experiments the authors should at least discuss the fact that what they are seeing in mice might not be reflective of what happens in primates, especially given the fact that the microbiome is mostly shaped by environmental factors that are dramatically different between primates and rodents.

---

## [Author Response]

Essential revisions:1. Please see the recommendations for authors for additional points of clarity. You're welcome to include additional data but textual edits are probably sufficient.2. Please also address the following additional points: How do you reconcile the fact that despite not expressing a-gal, apes and OWM are much more susceptible to sepsis as compared to species that do express a-gal, such as mice?

In response to this comment the following paragraph was added to the manuscript´s *Discussion:*

“When challenged experimentally with a bacterial inoculum or bacterial lipopolysaccharide (LPS), non-human primates appear to be far more susceptible to develop sepsis or septic shock, respectively, as compared to other mammalian lineages (Chen et al., 2019). […] This interpretation is consistent with resistance and disease tolerance to infection being negatively correlated (Raberg et al., 2007), such that traits increasing resistance might be associated with a reduction in disease tolerance, as an evolutionary trade-off.”

In the context of increased resistance to sepsis what seems to be more important – the remodeling of the microbiome by IgA or the increased effector function of IgG?

Our manuscript provides experimental evidence to suggest that antibodies targeting bacteria in the gut microbiota shape the microbiota composition of *Ggta1*-deficient mice in a manner that is distinct from antibodies generated by wild type mice. In doing so, antibodies generated by *Ggta1*-deficient mice reduce the microbiome pathogenicity, as assessed by development of lethal forms of sepsis upon systemic infection by microbiota inoculum. While the mechanism underlying microbiota shaping is antibody-dependent, the effector mechanism enhancing resistance against the shaped microbiota acts irrespectively of antibodies. We added the following paragraph to the manuscript's *Discussion* to highlight this point:

“We propose that *Ggta1* deletion in mice increases resistance to bacterial sepsis via two distinct antibody-dependent mechanisms. […] It is possible however, that similar to IgG(Singh et al., 2021), the absence of αGal from the glycan structures of different Ig isotypes, including IgA, modulates their effector function, when targeting immunogenic bacteria in the microbiota.“

The fact that the microbiome of laboratory mice and primates is likely very different and therefore further experiments are needed to evaluate if the findings described in this work directly translate to a mechanisms of increased resistant to sepsis in primates.

We addressed this comment by editing the Discussion section, as follows:

“While our studies do not provide direct evidence that the loss of αGal played a major role in shaping the microbiota composition of Old- *vs.* New World primates this notion is supported, indirectly, by the recent finding that mutations altering the expression of human ABO blood group glycans are associated with shaping of the bacterial composition of the gut microbiota (Rühlemann et al., 2021). […] Whether or not this is the case, considering that microbiota composition is shaped mostly by environmental factors, remains to be formally established.”

Please discuss the potential ramifications of IgG glycosylation changes and after α Gal loss. Given the demonstrated role of sialylation for effector function of IgG, could it be that lack of α Gal actually leads to higher levels of terminal sialic acid on IcG Fc N-glycans, and what would be the functional effects of that?

IgG carry biantennary glycan structures, N-linked to the Fc domain on an evolutionarily conserved asparagine (N297). These glycan structures contain varying amounts of N-acetylglucosamine, mannose, fucose, galactose and terminal sialic acids (Neu5Gc or Neu5Ac), which are linked to galactose (Cobb, 2019). When present in these glycan structures, αGal residues can modulate IgG-Fcγ binding and IgG effector function, as demonstrated in mice (Singh et al., 2021). This suggests that the loss *GGTA1* function during primate evolution might have exerted a similar effect on the human IgG effector function. Humans carry in addition, a non-functional CMP-N-acetylneuraminic acid hydroxylase (*CMAH*) gene and as such fail to produce terminal N-glycolylneuraminic acid (Neu5Gc) residues. Instead, human IgG-associated glycan strictures express terminal N-acetylneuraminic acid (Neu5Ac) residues, lacking the oxygen atom added by CMAH on the Neu5Gc residue. Terminal Neu5Ac residues in the biantennary glycan structures of IgG are linked to galactose and exert a major effect on IgG effector function (Cobb, 2019). Presumably, the combined evolutionary loss of terminal αGal and Neu5Gc residues from the biantennary glycan structures of human IgG is a defining feature of human IgG effector function. This remains however, to be formally established.

Reviewer #1 (Recommendations for the authors):CMAH/ Cmah loss-of function occurred at least 8 times concergently across mammals: Human, New-World monkey, Mustelids, twice in microbats, white tailed deer, sperm whale, and hedgehogs (Springer et al. 2014, PMID: 25124893, Ng et al. 2014 PMID: 25517696, Peri et al. 2018 PMID:29206915).

While natural selection and fixation of *GGTA1* loss-of-function mutations is a rare event, it probably occurred several times in non-mammalian vertebrates, as suggested by the apparent lack of αGal expression in cells isolated from fish, frogs, quails, ducks or chickens (Galili et al., 1988). While it is assumed that natural selection and fixation of *GGTA1* loss-of-function mutations has not occurred in mammals, perhaps other than Old-World primates, this remains to the best of our knowledge, to be formally established.

In stark contrast to α Gal, no microbe has yet been documented to be decorated with intrinsic N-glycolyl neuramninic acid, so microbiome immunization is very unlikely in that contsetx, although some human pathobionts, e.g. Hemophilus may accumulate it from the human host, where dietary Neu5Gc can become incorporated in human glycoconjugates.

Preformed αGal-specific antibodies are protective against zoonotic diseases caused by blood borne pathogens expressing αGal-like glycans, including membrane-enveloped viruses (Takeuchi et al., 1996), bacteria (Hamadeh et al., 1992) and protozoan parasites(Soares and Yilmaz, 2016; Yilmaz et al., 2014). This suggests that natural selection of *GGTA1* loss-of-function mutations was propelled by the emergence of protective immunity targeting pathogens expressing αGal-like glycans. The apparent absence of N-glycolylneuraminic acid expression in microbes would suggest that the process of natural selection of *CMAH* loss-of-function mutations was not propelled by the emergence of protective immunity against pathogens expressing this glycan. An alternative interpretation however, would be that loss of *CMAH* function and the emergence of antibodies directed against N-glycolylneuraminic acid might confer protective immunity against zoonotic membrane-enveloped viruses. In support of this notion, enveloped viruses can carry host cell derived glycans, suggesting that antibodies directed against N-glycolylneuraminic acid might act as an immediate resistance mechanism against zoonotic transmission from primates carrying a functional *CMAH* to humans carrying a non-functional *CMAH*. This hypothesis remains however, to the formally tested.

The use of Bandeiraea simplicifolia plant lectin (BSL-B4) for staining of α Gal glycan epitopes is well established. Nevertheless, using a negative control, such as treatment with coffee bean galactosidase specifically acting on terminal α 1,3 Galactose to confirm the α-Gal epitopes on various microbes might be very useful.

This is indeed a standard control for the specificity of the α-gal detection. We have not performed these experiments for logistic reasons.

The authors propose that the lack of GGTA in humans and other primates of Africa and Asia might have helped mitigate sepsis pathology, but they should address the well-known observation that humans are much more prone to sepsis than other mammals, INCLUDING Old World monkeys and apes.Many unusual things have happened to immunity on the lineage leading to modern humans. Could the authors discuss any evidence that other Old World Primates are somehow less prone to sepsis than α Gal positive mammas, including their New World primate cousins?

This is indeed an important point towards the validation of our findings in mice towards a better understanding of primate immunity and evolution. We are not aware of experimental evidence comparing the susceptibility to sepsis in Old *vs.* New World Primates. In response to this comment (also raised by Reviewer 3) we introduced the following paragraph in the Discussion section. Of note, we had previously discussed this specific point (Singh et al., 2021) and have done so here in a manner that is, we believe, non-redundant.

“When challenged experimentally with a bacterial inoculum or bacterial lipopolysaccharide (LPS), non-human primates appear to be far more susceptible to develop sepsis or septic shock, respectively, as compared to other mammalian lineages (Chen et al., 2019). […] This interpretation is consistent with resistance and disease tolerance to infection being negatively correlated (Raberg et al., 2007), such that traits increasing resistance might be associated with a reduction in disease tolerance, as an evolutionary trade-off.”

Page 2 Line 72 IgA limits, not limit.

This was corrected, thank you for pointing it out.

Reviewer #3 (Recommendations for the authors):One major open question is how much of the phenotypes described herein are representative of what happens in primates once they lost GGTA1 (which is ultimately what motivated the work ). Although I appreciate that similar experiments cannot be performed in primates there are a few experiments that would be doable. For example, the authors could analyze microbiome data from New- vs Old-world primates (humans included) to see if the former (who still express a-gal) do indeed have a less pathogenic microbiome (there is data out there already available). Another experiment would be to perform the sepsis challenges using cecal inoculum derived from New- vs Old-world primates/humans and see if the microbiome from New world monkeys is indeed less pathogenic than that from Old-world primates, as predicted by their model. In the absence of doing such experiments the authors should at least discuss the fact that what they are seeing in mice might not be reflective of what happens in primates, especially given the fact that the microbiome is mostly shaped by environmental factors that are dramatically different between primates and rodents.

We agree with this comment and have addressed it, as suggested, by editing the Discussion section, as follows:

“While our studies do not provide direct evidence that the loss of αGal played a major role in shaping the microbiota composition of Old- *vs.* New World primates this notion is supported, indirectly, by the recent finding that mutations altering the expression of human ABO blood group glycans are associated with shaping of the bacterial composition of the gut microbiota (Rühlemann et al., 2021). […] Whether or not this is the case, considering that microbiota composition is shaped mostly by environmental factors, remains to be formally established.”

References:

Ayres, J.S. (2016). Cooperative Microbial Tolerance Behaviors in Host-Microbiota Mutualism. Cell 165, 1323-1331.

Chen, L., Welty-Wolf, K.E., and Kraft, B.D. (2019). Nonhuman primate species as models of human bacterial sepsis. Lab Anim (NY) 48, 57-65.

Cobb, B.A. (2019). The history of IgG glycosylation and where we are now. Glycobiology 30, 202-213.

Galili, U., Shohet, S.B., Kobrin, E., Stults, C.L., and Macher, B.A. (1988). Man, apes, and Old World monkeys differ from other mammals in the expression of α-galactosyl epitopes on nucleated cells. J Biol Chem 263, 17755-17762.

Hamadeh, R.M., Jarvis, G.A., Galili, U., Mandrell, R.E., Zhou, P., and Griffiss, J.M. (1992). Human natural anti-Gal IgG regulates alternative complement pathway activation on bacterial surfaces. J Clin Invest 89, 1223-1235.

Martins, R., Carlos, A.R., Braza, F., Thompson, J.A., Bastos-Amador, P., Ramos, S., and Soares, M.P. (2019). Disease Tolerance as an Inherent Component of Immunity. Annual Reviews of Immunology 37.

Medzhitov, R., Schneider, D., and Soares, M. (2012). Disease Tolerance as a Defense Strategy. Science 335, 936-941.

Olson, M.V. (1999). When less is more: gene loss as an engine of evolutionary change. Am J Hum Genet 64, 18-23.

Raberg, L., Sim, D., and Read, A.F. (2007). Disentangling genetic variation for resistance and tolerance to infectious diseases in animals. Science 318, 812-814.

Rühlemann, M.C., Hermes, B.M., Bang, C., Doms, S., Moitinho-Silva, L., Thingholm, L.B., Frost, F., Degenhardt, F., Wittig, M., Kässens, J., et al. (2021). Genome-wide association study in 8,956 German individuals identifies influence of ABO histo-blood groups on gut microbiome. Nature Genetics 53, 147-155.

Singh, S., Thompson, J.A., Yilmaz, B., Li, H., Weis, S., Sobral, D., Truglio, M., Aires da Silva, F., Aguiar, S., Carlos, A.R., et al. (2021). Loss of α-gal during primate evolution enhanced antibody-effector function and resistance to bacterial sepsis. Cell Host Microbe 29, 347-361.e312.

Soares, M.P., and Yilmaz, B. (2016). Microbiota Control of Malaria Transmission. Trends Parasitol 32, 120-130.

Takeuchi, Y., Porter, C.D., Strahan, K.M., Preece, A.F., Gustafsson, K., Cosset, F.L., Weiss, R.A., and Collins, M.K. (1996). Sensitization of cells and retroviruses to human serum by (α 1-3) galactosyltransferase. Nature 379, 85-88.

Vonaesch, P., Anderson, M., and Sansonetti, P.J. (2018). Pathogens, microbiome and the host: emergence of the ecological Koch's postulates. FEMS microbiology reviews 42, 273-292.

Wang, X., Grus, W.E., and Zhang, J. (2006). Gene losses during human origins. PLoS Biol 4, e52.

Yilmaz, B., Portugal, S., Tran, T.M., Gozzelino, R., Ramos, S., Gomes, J., Regalado, A., Cowan, P.J., d'Apice, A.J., Chong, A.S., et al. (2014). Gut Microbiota Elicits a Protective Immune Response against Malaria Transmission. Cell 159, 1277-1289.